# Post-drought decline of the Amazon carbon sink

Yan Yang[1,2,3], Sassan S. Saatchi[1,3], Liang Xu [1,3], Yifan Yu[3], Sungho Choi [2], Nathan Phillips[2], Robert Kennedy[4], Michael Keller[3,5], Yuri Knyazikhin[2] & Ranga B. Myneni[2]

Amazon forests have experienced frequent and severe droughts in the past two decades. However, little is known about the large-scale legacy of droughts on carbon stocks and dynamics of forests. Using systematic sampling of forest structure measured by LiDAR waveforms from 2003 to 2008, here we show a significant loss of carbon over the entire Amazon basin at a rate of $0.3 \pm 0.2$ (95% CI) PgC yr$^{-1}$ after the 2005 mega-drought, which continued persistently over the next 3 years (2005–2008). The changes in forest structure, captured by average LiDAR forest height and converted to above ground biomass carbon density, show an average loss of $2.35 \pm 1.80$ MgC ha$^{-1}$ a year after (2006) in the epicenter of the drought. With more frequent droughts expected in future, forests of Amazon may lose their role as a robust sink of carbon, leading to a significant positive climate feedback and exacerbating warming trends.

[1] Institute of Environment and Sustainability, University of California, Los Angeles, CA, USA. [2] Department of Earth and Environment, Boston University, Boston, MA, USA. [3] Jet Propulsion Laboratory, California Institute of Technology, Pasadena, CA, USA. [4] Dept. of Earth, Ocean, and Atmospheric Sciences, Oregon State University, Corvallis, OR, USA. [5] Int. Institute of Tropical Forestry & Int. Programs, USDA Forest Service, Washington, USA. Correspondence and requests for materials should be addressed to Y.Y. (email: yangyannn@gmail.com)

Amazon forests contain nearly half of the tropical forest carbon stocks[1] and play a major but uncertain role in the global carbon budget[2–5]. In the past two decades (1998–Present), Amazon forests have experienced frequent and severe droughts resulting from climate variability at approximately 5–6 year intervals, starting with the 1998–99 El Nino, extreme water deficits in 2005 and 2010 resulting from the warming anomaly of Tropical North Atlantic (TNA)[6–8], and the recent 2015–16 El Nino[9]. Impacts of droughts on carbon dynamics of forests of Amazon have been recorded in terms of short-term (1–3 years) tree mortality and biomass loss from small-scale observations in inventory plots[10–13]. Repeated measurements from inventory plots show a significant legacy effect after the 2005 drought[14], with increasing tree mortality and carbon loss, temporarily converting Amazon forests from a net sink[15] of about 0.71 MgC ha$^{-1}$ yr$^{-1}$ to a net source of carbon to the atmosphere of about 5.3 MgC ha$^{-1}$ for forest subjected to a 100 mm increase in water deficit[12]. However, extrapolations from plot-level studies to the entire Amazon region may have large uncertainty due to variability of forest composition and the climate and edaphic conditions controlling the forest function and resilience to climatic stress[16]. The ability of land-surface models, including dynamic global vegetation models (DGVMs), to project the broad-scale effects of climate extremes on Amazon forest C stocks and dynamics requires initialization with data on the spatial heterogeneity of forest biomass and productivity across the landscape, but such models currently are limited by many factors[17,18], including a general lack of realistic tree mortality functions[19] and uncertainties associated with plant physiological responses to $CO_2$ enrichment[20]. Therefore, the large-scale effects of droughts, and their legacy, over forests of the Amazon region remain uncertain. If tree mortality and disturbance of forest productivity observed in plots are widespread, the carbon loss from droughts will be significant and may have adverse consequences for global carbon cycle and its feedbacks to climate[12,21].

In this paper, we analyze LiDAR measurements of forest structure, systematically sampled by the Geoscience Laser Altimeter System (GLAS) aboard the Ice, Cloud, and the Elevation Satellite (ICESat) from 2003 to 2008 to quantify the changes in structure and carbon stocks of forests as a result of the 2005

drought. LiDAR samples of vertical structure of forests are recognized as the most effective remote sensing approach to quantify the above ground forest biomass[1,22–26]. We examine whether there have been widespread changes of forest structure from tree mortality or canopy disturbance either in the western Amazon (4°S–12°S, 76°W–66°W), where the 2005 drought impacts were severe, or the entire Amazon (19°S – 12°N; 81° W–44°W) that experienced water deficit and temperature anomaly. The analyses are focused on a 6-year period (2003–2008) observations of intact forests to assess whether the 2005 drought had a legacy effect that extended spatially and temporally beyond its occurrence. By converting the LiDAR measurements to forest above ground, and through allometry to below-ground carbon density, we quantify the net carbon balance of the Amazon forests and its attributions into sources and sinks of carbon during the observational period. Our results demonstrate the widespread and persistent effects of episodic droughts on carbon dynamics of the Amazonian forests and its significant post-drought impacts on the global carbon sources and sinks.

## Results

**Changes in forest structure.** We stratified the Amazon forests into five regions based on the level of cumulative water deficit (CWD) anomaly for the months of July, August, and September (JAS) in 2005 calculated from 10 years (2000–2009) satellite rainfall data from Tropical Rainfall Measuring Mission (TRMM) combined with in situ meteorological networks of observations (see Methods) (Fig. 1a). The stratification provided a gradient of drought impacts in 2005, separating the extreme drought (ED) of the west from the severe drought (SD) of the southwest, the relatively moderate drought (MD) of the south, light drought (LD) of the northeast, and areas of almost no water deficit (ND) in the northwest Amazon. The stratification includes only pixels (5 km × 5 km) falling in the humid tropical forest category with more than 60% tree cover, and forest changes refer only to those focuses on intact forests with less than 1% deforestation and fire events happened in each pixel (see Methods).

We observed a widespread decline of forest canopy height after the 2005 drought over three drought-impacted (ED, SD, and MD) regions (Fig. 1b). The height measurement was based on the 90th

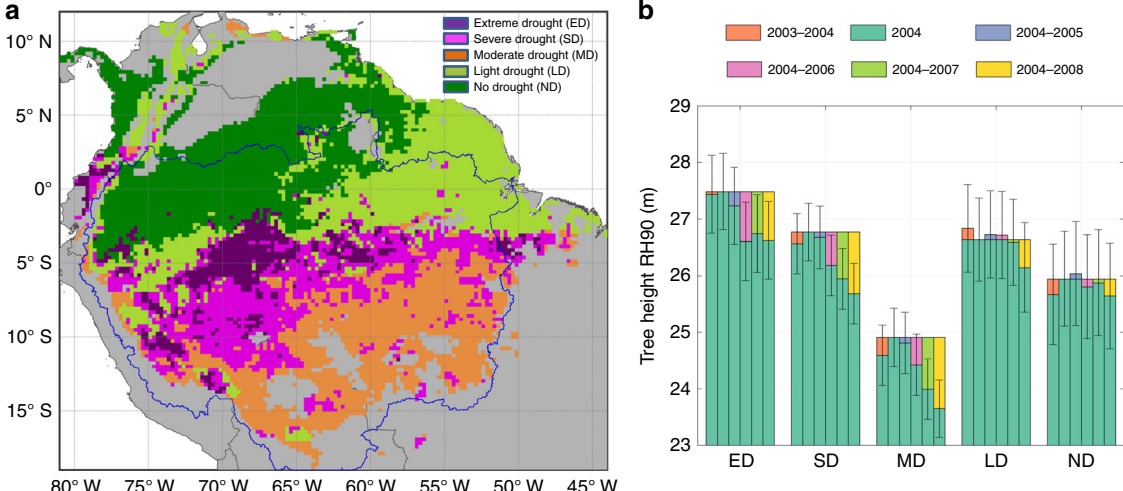

**Fig. 1** Spatial patterns of rainfall and corresponding canopy structure changes in each climatic region. **a** Drought classification map derived from rainfall data[82]. **b** Interannual changes of top canopy structure (RH90) relative to RH90 in 2004 for each climatic region defined in panel **a**. The abbreviations ED, SD, MD, LD, and ND are regions of extreme drought, severe drought, moderate drought, light drought, and no drought. The purple line in panel **a** delineates the boundary of Amazon basin area. Non-tropical forests in panel **a** were colored in gray. The error bars in panel **b** stand for 95% confidence intervals. Estimations in panel **b** were derived from the spatial modeling with filled data gaps (SMF) method (see Methods)

percentile of return energy (RH90) of GLAS LiDAR waveforms, capturing the variation of forest structure and the upper canopy volume and gaps, and collected at the end of dry season (Oct–Nov) for each year (see Methods). The decline of this height metric (~24 m on average in the Amazon) represented the impacts of the disturbance (tree falls, defoliation, canopy damage) on the forest above ground biomass (AGB). The most significant decline happened in the southwest (ED)—the epicenter of the 2005 drought, with RH90 declining by 0.88 ± 0.69 m one year after the drought, indicating loss or disturbance (e.g., defoliation or tree fall gaps) of canopy trees (Supplementary Table 1). The SD region also experienced a decline in forest height (by 0.59 ± 0.53 m), but the magnitude was comparable to the ED region only 2 years after the drought (0.82 ± 0.54 m decline from 2004 to 2007). The MD region also suffered from seasonal water deficits and showed a relatively steady decline although the change of forest structure was not significant (0.49 ± 0.54 m) until 2007 (0.92 ± 0.54 m). This consistent decline of canopy height after the drought event was significantly higher than the variance associated with the spatial variability of forest structure (if treating the change as independent events, we have the uncertainty of change $\sigma = \sqrt{\sigma_1^2 + \sigma_2^2}$, where $\sigma_1$ and $\sigma_2$ are the uncertainty associated with structure in year 1 and year 2). The MD region was also impacted by severe logging activities in the Brazilian Amazon and the decline of forest structure may also be attributed to any post-drought increase in mortality of trees in logged forests (In contrast, the LD and ND regions have much less variation in forest height from 2004 to 2008, and neither of the changes are statistically significant (−0.49 ± 0.79 m for LD and −0.30 ± 0.93 m for ND regions). Although changes of canopy height were not statistically significant within the first year after the drought, all regions except LD and ND experienced significant declines of forest height after 3 years. Throughout the observational period (through the end of 2008), the average canopy height did not show any obvious sign of recovery, indicating at least a 3–4-year legacy of the 2005 drought,

potentially combined with the stress from increasing land-surface temperature[27].

**Biomass and carbon changes**. Using existing established models[1,5], we further converted the GLAS LiDAR samples for each year to AGB and total carbon by adding the below ground biomass (BGB) using tree allometry (see Methods). The regional AGB changes in the western and southern Amazon showed significant losses of biomass after the 2005 drought (Fig. 2a). Specifically, ED and SD regions had losses of biomass 1 year after the 2005 drought, but the MD region experienced a slower decline of biomass to become significant only after 3–4 years (2004–2008). Regions LD and ND, with plenty of rainfall throughout the observational period, did not have a significant water stress and decline in AGB (and no significant decline in total carbon). Combining regions LD and ND as North, and regions ED, SD, and MD as South, we found clearly distinguishable patterns of total carbon changes between North and South (Fig. 2b, c). Forests in the Northern Amazon remained relatively unchanged on average, but Southern Amazon forests declined after the drought event. Total carbon over the entire Amazon basin area also showed a steady decline from the end of 2004 (Fig. 2d) with no sign of recovery. The average loss of carbon across the entire Amazon basin was 0.27 ± 0.15 PgC yr⁻¹ (Table 1). While the uncertainty of this estimate precludes evaluation of biomass decline immediately after the drought event, the lagged effect and the prolonged impact of the drought enabled us to find a statistically significant estimate of biomass loss starting soon after the drought.

For the period of the study, we assume gross carbon emissions from forests of the Amazon Basin are from a combination of three sources: deforestation including fires from slash and burn clearing, closed-canopy forest fires not accounted in deforestation, and the drought-related disturbance impacting the intact old growth forests. Here, we ignore the emissions from logging,

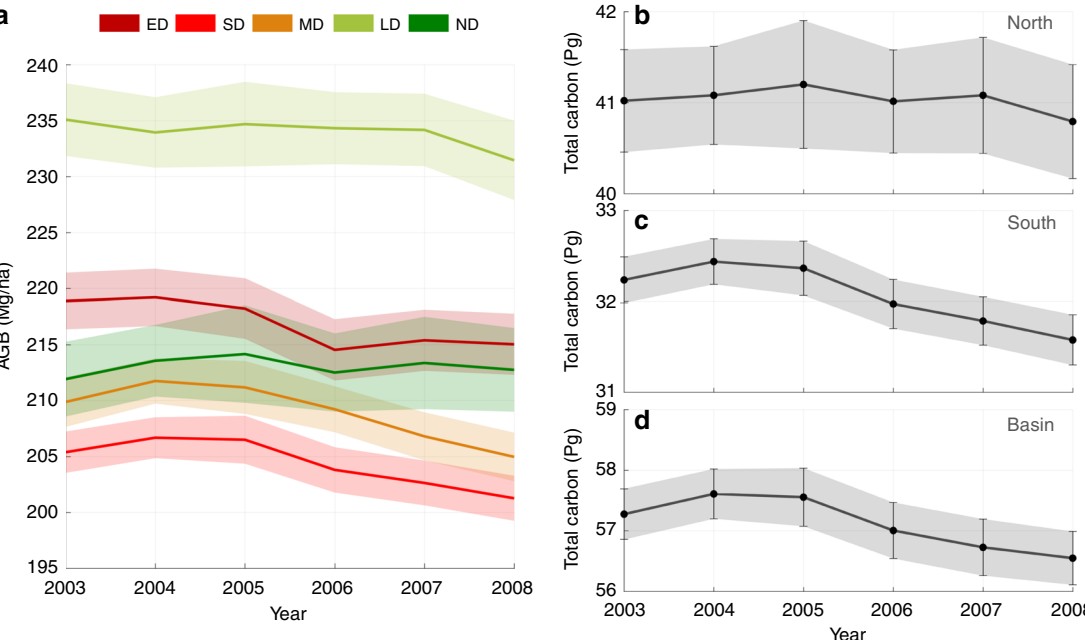

**Fig. 2** GLAS-derived biomass changes in Amazon. **a** Interannual above ground biomass (AGB) changes of regional means in five climatic regions (see Fig. 1a); **b** Interannual total carbon changes in North; **c** Interannual total carbon changes in South; **d** Interannual total carbon changes over the Amazon basin. The error bars (shaded area) stand for 95% confidence intervals. We combine regions LD and ND to be North, and regions ED, SD, and MD as South. The abbreviations ED, SD, MD, LD, and ND are regions of extreme drought, severe drought, moderate drought, light drought and no drought. All estimations in this figure were derived from the spatial modeling with filled data gaps (SMF) method (see Methods)

**Table 1 Detected changes of Amazon forests from 2004 to 2008**

| Regions/Indicators | 2004-2008 | Annual Change | Annual % Change |
|---|---|---|---|
| RH90(m, South) | $-1.11 \pm 0.48$[a] | $-0.28 \pm 0.12$[a] | $-1.06 \pm 0.45\%$[a] |
| RH90(m, North) | $-0.39 \pm 0.82$ | $-0.10 \pm 0.20$ | $-0.37 \pm 0.78\%$ |
| RH90(m, Basin) | $-0.82 \pm 0.45$[a] | $-0.20 \pm 0.11$[a] | $-0.77 \pm 0.43\%$[a] |
| AGB (Mg ha$^{-1}$; South) | $-5.72 \pm 2.52$[a] | $-1.43 \pm 0.63$[a] | $-0.68 \pm 0.30\%$[a] |
| AGB (Mg ha$^{-1}$; North) | $-1.60 \pm 4.58$ | $-0.40 \pm 1.15$ | $-0.18 \pm 0.51\%$ |
| AGB (Mg ha$^{-1}$; Basin) | $-4.10 \pm 2.32$[a] | $-1.03 \pm 0.58$[a] | $-0.47 \pm 0.27\%$[a] |
| Total Carbon (PgC; South) | $-0.86 \pm 0.38$[a] | $-0.22 \pm 0.09$[a] | $-0.66 \pm 0.29\%$[a] |
| Total Carbon (PgC; North) | $-0.29 \pm 0.83$ | $-0.07 \pm 0.21$ | $-0.18 \pm 0.50\%$ |
| Total Carbon (PgC; Basin) | $-1.06 \pm 0.60$[a] | $-0.27 \pm 0.15$[a] | $-0.46 \pm 0.26\%$[a] |

Gross changes are shown for GLAS-derived top canopy height (RH90) and carbon storage (average AGB and Total Carbon) in the north, south, and the entire Amazon Basin. The uncertainty values added to the mean changes are at 95% confidence intervals. Annual change is the average change per year calculated from 2004 to 2008, and the % change is the relative change of each region to the observations in 2004. Region South (shown in Fig. 3a) combines ED, SD, and MD (Fig. 1a), while region North combines LD and ND. The Basin region is delineated in Fig. 1a with purple lines
[a]The change is significant at 5% level

understory fires, and any other cryptic degradation processes as they may not be detected by remote sensing observations, assuming the carbon loss is small (less than 10%)[28,29] and the carbon gain from subsequent regeneration from deforestation and logging is negligible and have no significant annual trends during the period of our study[30]. From satellite data (see Methods), we identified forest pixels with fire and deforestation activities. Most of fires during our study period happened in the southern Amazon, near the edges of tropical and transitional forests (Fig. 3a). A large fraction of deforestation events was also included in the fire pixels, but there were other fire impacted pixels in the north and western Amazon that were unrelated to deforestation and were treated separately. The net change of carbon in intact forests was calculated in each region by eliminating all pixels (~25 km$^2$) with more than 1% presence of deforestation and fires detected by satellite imagery, and hence excluding most of the forest edges that may have been impacted by understory fires undetected in satellite products[31]. Deforestation had a stable contribution of around 0.15 PgC yr$^{-1}$ to the total emission (0.12 PgC from the south and 0.03 PgC from north) (Fig. 3b, c: sinks are negative and sources are positive) each year from 2003 to 2008. Fire contributed slightly less (0.11 PgC yr$^{-1}$) to carbon emissions (0.09 PgC from the South and 0.02 PgC from North) mostly due to the smaller emission factor[32,33]. Spatially, the northern Amazon had fewer deforestation and fire activities (Fig. 3a, b), which explains the much larger emissions found in the South (Fig. 3c). Excluding the impact of fire and deforestation, the net carbon change in remaining intact forests were statistically unchanged in the North (Fig. 3b), but switched from an insignificant negative (sink) of approximately 0.2 PgC before the drought to a significant source of 0.4 PgC a year after the drought (Fig. 3c), and about 0.9 PgC 3 years after (Fig. 3d). The year-to-year variations of the forest carbon change in the South were all pointing to a source of carbon to the atmosphere, and importantly, the contribution was statistically significant when the disturbance continued, suggesting a strong impact of the drought legacy on the carbon cycling of intact old growth forests of the Amazon. Although our year-to-year estimate of the magnitude of carbon sources and sinks in the Amazon may still have some residual effects from potential forest degradation or some understory fires[31], the long-term trend and the gradual decline of the carbon stock to a net source are mainly associated with the 2005 drought and in agreement with measurements from plot networks[11].

## Discussion
The annual LiDAR footprints are statistical samples (without repeated measurements of the same footprint over time) of forest canopy height, capturing the average state of the forest structure over each region. The LiDAR RH90 metric provides the approximate mean top canopy height and its value declines if the canopy is disturbed due to defoliation, structural damage, and tree fall, or any increases from growth and gap filling (Supplementary Fig. 1). For old growth forests under steady-state conditions, the year-to-year average changes of LiDAR measurements over a large region should be negligible and only represent the background disturbance and recovery processes[34]. Therefore, the decline of average RH90 in ED and SD regions suggests that many large canopy trees have been impacted significantly, consistent with theoretical and empirical evidence that larger trees are most affected by ED events[35,36]. By choosing the LiDAR samples from the late dry season for each year (see Methods), we also expect the signal of height decline is independent of any potential effect of seasonal leaf phenology[37].

The gross decline of forest biomass after the 2005 drought from LiDAR analysis may underestimate the total loss of biomass compared with ground inventory data[11,12]. In LiDAR analysis, the reduction of the tree height due to increasing gaps from tree falls and defoliation of dead trees have been converted to biomass loss. Whereas, in ground inventory, dead trees are removed from biomass calculations for gross committed emissions but allowed to decompose over long period (~30 years) for net carbon emissions[29,30]. In our study, the LiDAR derived changes of biomass can be dominated by the old-growth intact forests due to the higher weight associated with the RH90 metric (see Methods), but there was also understory forest growth (Supplementary Fig. 2), partly reflected in our RH30 metric used in the allometry. Furthermore, the annual changes of forest carbon stocks from non-overlapping LiDAR samples have larger uncertainty than repeated LiDAR measurements or inventory plots, causing difficulty in attributing year-to-year decline of forest carbon in the South as statistically significant net source.

Our estimates of carbon loss as a consequence of the 2005 drought must be regarded as the decline of the forest carbon sink and not the absolute magnitude of the sink. However, the persistence of the carbon loss few years after the drought point to gradual and longer impact of episodic droughts on the Amazon forest. Severe episodic droughts in the Amazon have been recorded in the last decade (2005, 2010, and 2015) and are expected to be more frequent in the future[9]. The pervasive drought legacies in these ecosystems[14] may have long-term effects on the tropical carbon sink and the overall terrestrial carbon budget, leading to an accelerated positive feedback to regional and global climate. The repeated sampling of LiDAR data enables documenting post-drought structural changes and carbon losses from the entire Amazon, corroborating what was found at the smaller scales in research plots that documented the increase in tree mortality and potential decline of tree productivity[38]. Our

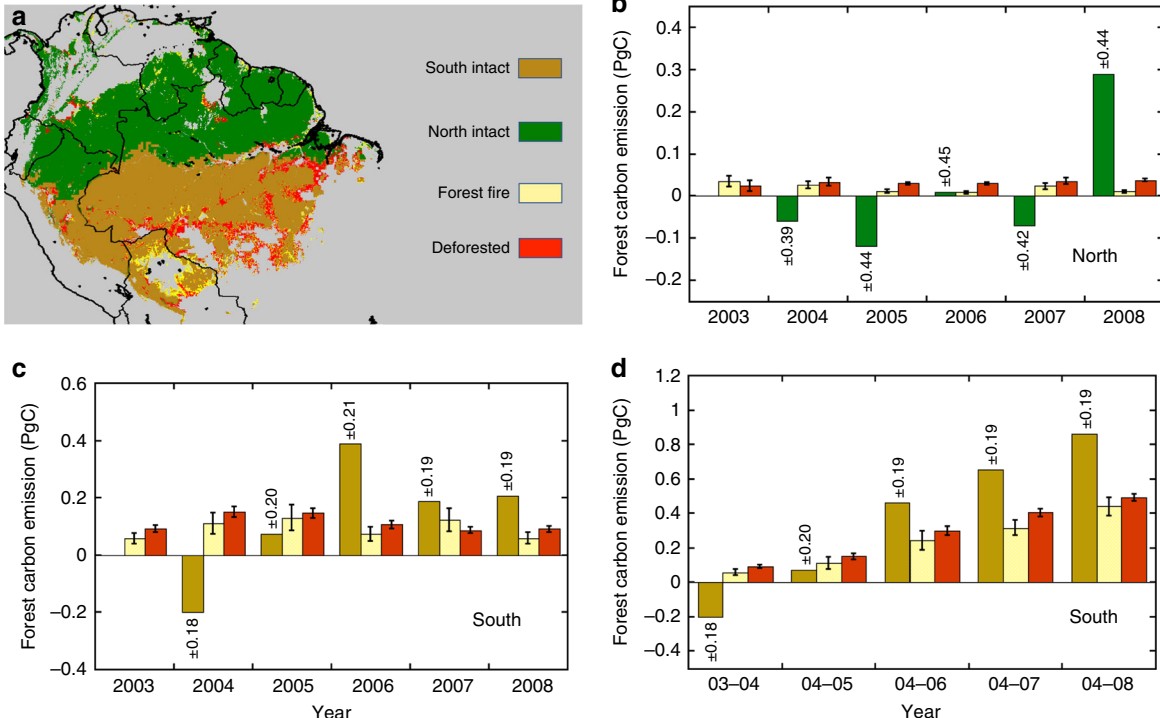

**Fig. 3** Net carbon change and emissions from fire and deforestation. **a** Classification map showing pixels with intact Amazon forests belonging to South and North, as well as fire and deforestation activities in tropical forests. **b** Annual contributions of fire and deforestation emissions compared to the net forest carbon change in the North region. **c** Annual contributions of fire and deforestation emissions compared to the net carbon change in the South region. **d** Cumulative contributions of net forest carbon change, fire and deforestation emissions in the South relative to the 2004 level. The colormap in panel **a** was generated using rainfall[82], MODIS fire[50] and Landsat[51] products. Pixels were identified as fire or deforested when at least 1% of these land-use activities were detected during our observational period. In panels **b-d**, the positive sign indicates carbon emission to the atmosphere, and negative emission refers to carbon sink. The error bars in estimating emissions are at 95% confidence intervals

results clearly indicate that the Amazon forests may lose their role as a robust sink of atmospheric carbon in the face of repeated severe droughts[4,39]. With detailed eco-hydrological studies of forest function from a combination of widespread ground plots[13] and repeated observations from space[7,40], the underlying causes of these changes and their spatial extent and long-term effects can be explored with less uncertainty in future.

## Methods

**Remote sensing data**. Our study region covers the entire Amazon forests within the boundary of north and central South America (19°S–12°N; 81°W–44°W). We used pixels identified as Evergreen Broadleaf Forests (EBF) in the latest Moderate Resolution Imaging Spectroradiometer (MODIS) Land Cover (LC) product[41]. The EBF pixels were defined from the year-2005 LC map for our observational period (2003–2008) to ensure capturing the forest changes triggered by the 2005 Amazon drought. The MODIS Vegetation Continuous Field (VCF) product[42] was also used to further stratify the pixels. By taking the maximum VCF values from 2003 to 2008 as $VCF_{max}$, our study region would only focus on the dense tropical forests ($VCF_{max} > 60\%$). The threshold of VCF will exclude a large number of partially forested areas across the arc of deforestation.

The centerpiece of datasets used in this study is the spaceborne GLAS Lidar waveform measurements. GLAS sensor aboard the Ice, Cloud and land Elevation Satellite (ICESat) is the first spaceborne waveform sampling Lidar instrument for continuous global observation of the Earth. It emits short duration laser pulses and records the echoes reflected from the Earth's surface[43]. For vegetated surfaces, the return echoes or waveforms are the function of canopy vertical distribution and ground elevation within the area illuminated by the laser (the footprint), thus reflecting the canopy structure information[1,44,45]. Here, we used the GLAS/ICESat L2 Global Land Surface Altimetry Data (GLAH14) product and filtered the original data using a series of stringent quality controls and processing steps (see GLAS preprocessing). We calculated the canopy height metrics from reconstructed waveform data to study the interannual changes over the retrieval period (2003–2008).

Other ancillary data, including the radar backscatter from the QuickSCAT satellite[46] (QSCAT), and the MODIS Multi-Angle Implementation of Atmospheric

Correction Algorithm (MAIAC) EVI product[47], can directly or indirectly capture the structural and carbon changes in the Amazon forests[7,48,49]. Together with the fixed MODIS $VCF_{max}$ layer to account for the effective canopy cover, we used these spatially and/or temporally continuous satellite data to interpolate Lidar samplings, so as to create the Lidar-based mapping for each year. We further explored the fire frequencies from the MODIS Burned Area Product[50] to identify regions with forest fires. Other activities causing forest cover loss, such as deforestation, were analyzed using Global Forest Cover (GFC) loss event data derived from Landsat imagery[51]. We also categorized the tropical climate in the Amazon Basin using rainfall data from TRMM 3B43 product[52]. The 3B43 product combines rainfall estimates from TRMM and other satellites, as well as the global gridded rain gauge data, and provides the monthly precipitation rate at $0.25° \times 0.25°$ spatial resolution starting from 1998[53]. The last dataset we used in our study was a benchmark biomass map for tropical forests[1]. Using the benchmark biomass map as a reference, we interpreted the GLAS height metrics into changes of carbon storage over the Amazon forests. However, the original biomass map and its spatial variation did not directly impact our results.

**GLAS preprocessing**. The GLAS GLAH14 product is a land product containing the land elevation and elevation distributions[54]. Within our study region, we have a total of 7.5 million GLAS shots in the format of GLAH14 for the study period from 2003 to 2008 (Supplementary Table 2). But not all data are useful to study the interannual changes, and thus data screening is necessary. To get an unbiased estimation of canopy structure from the original data product, we performed the following necessary data preprocessing steps (Supplementary Table 3): a LC filter, a VCF filter, a seasonal filter, a saturation filter, a 2-peak filter, a cloud filter, and a slope filter.

The MODIS LC map in 2005 defines the regions where tropical forests are located. We used this map to keep GLAS shots located only in these forested pixels. The VCF filter is an additional LC-based data screening step to focus our study area only on those dense forests. We ruled out all GLAS shots located in pixels with less than 60% tree cover. The percent tree cover data were extracted from the MODIS $VCF_{max}$ extracted from the max VCF between 2003 and 2008. The Amazon forests, though considered evergreen, have seasonal variations due to climate patterns and regional differences[55,56], as well as the canopy structure and variations in leaf optics that can impact the photosynthetic capacities and carbon exchanges[37,49,57]. To remove the potential seasonal effects of GLAS data, we

checked the operational periods of GLAS[58] and used GLAS laser shots in October and November for our study from 2003 to 2008. This period corresponds to the end of dry and the beginning of the wet season in most of Amazon and has consistently larger number of samples from 2003 to 2008 compared to April–June period as the end of the wet season and the beginning of dry season. Lidar waveforms captured by the GLAS instrument may have pulse distortions when the received energy exceeds the linear dynamic range of GLAS detector. This happens often in areas with flat and bright surfaces[43]. Saturated return signals in forests can barely preserve the shape of waveform reflected from scattering elements within canopy. In this study, we removed the saturated GLAS shots by investigating the Saturation Correction Flag as a quality assurance (QA) step. GLAS returns from forests are different from the returns on ice sheets or bare ground surfaces, as the waveforms are often bi-modal or multimodal[45,59] caused by the time differences of separated returns from forest canopy and the underlying ground. GLAH14 product parameterizes the return waveforms into six Gaussian fits[60] and reduces the stored waveform information to merely 18 Gaussian parameters. To find the peaks (local maxima) in the GLAS returns, we reconstructed the waveforms from Gaussian parameters and removed observations with only one-peak return. This step ensures that the remaining shots have at least two scattering centers at different elevations. An additional check along the waveform was to ensure no obvious data gaps (zero returns for more than 1 meters) between peaks, as anomalous peaks might be captured by the sensor above or below the vegetation canopy. We also filtered out waveforms with ground return peak <0.2V, as it provides the best distinction between returns representing ground only and mixed signal returns from ground and vegetation[61]. GLAH14 product has a set of quality flags documenting the atmospheric conditions during the waveform retrieval[54]. The Atmosphere Characterization Flag (atm_char_flag) contains records of the atmospheric condition at the 1 Hz rate. We picked the 40 Hz GLAS laser shots only when atm_char_flag equals to 0 (clear sky). We calculated the terrain slope from at each GLAS shot by fitting the ground waveform into the Gaussian function[61]. To avoid the false detection of ground and the mixture of signals from both canopy and ground, we filtered all data with calculated slopes larger than 10°.

For each valid waveform, we reconstructed the return at 0.2 m interval by summing up the six Gaussian fits:

$$f(z) = \sum_{i=1}^{6} A_i e^{-(x-\mu_i)^2/2\sigma_i^2}, \tag{1}$$

where $A_i$, $\mu_i$, and $\sigma_i$ are the Gaussian parameters stored in the GLAH14 product, indicating the Gaussian amplitude, peak position, and standard deviation, respectively[62]. At this stage, external sources such as the terrain data from the Shuttle Radar Topography Mission (SRTM) can provide additional information needed to extract the ground return information for slope estimation and correction[63–65]. But the extraction of terrain information using external sources has difficulties in the dense tropical forests due to the shallow penetration of high-frequency radar interferometry (SRTM) that captures the scattering centers mostly in the upper canopies of closed forests[45,66]. Accounting for other uncertainties from spatial resolutions and geolocation errors, we applied the independent slope method (ISM) by estimating the terrain slope from the GLAS waveform[61] at each footprint location.

The 2-peak filter (5th step of GLAS data screening) reconstructs the waveform and finds the ground returns. The concept of ISM[61] is to fit the lowest waveform peak as a Gaussian function and set the width of the Gaussian fit ($W_{G_f}$) as the elevation range of ground (Supplementary Fig. 3). Knowing the mean footprint diameter ($\bar{D}$) as the average of major and minor axis lengths of the footprint ellipse, we calculated the slope as

$$\text{Slope} = \frac{W_{G_f} - W_m}{\bar{D}}, \tag{2}$$

where $W_m$ is the minimum width of the GLAS backscatter from a flat surface. The value of $W_m$ reflects the duration of transmitted signal and associated attenuation from scattering elements. Using the top 0.1 percentile data with the least $W_{G_f}$ for different amplitude intervals, we empirically built the linear relationship between $W_m$ (in meters) and the ground peak amplitude $A_g$ (in V) for the tropical forests of Amazon:

$$W_m = 2.04 + 0.40 \times A_g. \tag{3}$$

The final dataset therefore keeps only observations on flat terrain with slopes no greater than 10°. Using the lowest peak found during the amplitude filter step as the ground position, we derived the relative height metrics (RH) from GLAS waveforms by defining the RH positions corresponding to the 10th, 20th, …, and 90th percentile of waveform energy[22,67], and denoted them as RH10, RH20, …, and RH90.

**Sampling strategy**. The sampling nature of the GLAS instrument forms a spatial–temporal distribution of the forest height (or derived metrics such as biomass and carbon density). Because of the existence of spatial autocorrelation, spatial samplings of GLAS data are distinct from classical statistics. Conventional

random sampling draws the samples independently with an equal probability from the population. The population mean can thus be estimated from the sample arithmetic mean ($\bar{y} = \frac{1}{n}\sum_{i=1}^{n} y_i$) of samples and the variance of estimated mean is proportional to the sample variance normalized by the sample size ($V(\bar{y}) = \frac{\sigma^2}{n}$)[68]. In spatial sampling, observations are associated with geographic locations ($y(\mathbf{X}_1), y(\mathbf{X}_2),\ldots,y(\mathbf{X_n})$). The repeated exhaustive samplings can detect the temporal changes of regional mean or total quantities, such as the wall-to-wall maps derived from remote sensing data. When the wall-to-wall mapping is not available, we need appropriate sampling techniques to get unbiased estimations of regional quantities[69–72]. We tested three statistical methods including design-based and model-based sampling techniques to study the interannual changes of GLAS observations in the Amazon.

**Stratified random sampling**. Stratified random sampling (SRS) is a design-based sampling strategy that first divides the area into $K$ non-overlapping strata, and then selects spatially random samples from each stratum. The SRS method is generally more efficient than simple random sampling, while keeps the design-unbiased estimates of the population, unlike in the systematic sampling[68,69,73]. The raw GLAS samples are usually clustered, and the simple arithmetic mean of all samples could lead to a biased result. We selected a subset of the GLAS shots to form a spatially balanced point patterns so that conventional statistics can be applied. The regional mean of the SRS method is

$$\bar{y}_{str} = \Sigma_{k=1}^{K} w_k \bar{y}_k, \tag{4}$$

where $w_k$ is the weight of the $k$th stratum, and in our study is proportional to the dense forested area of each stratum. The variance of the estimated mean is

$$V(\bar{y}_{str}) = \Sigma_{k=1}^{K} w_k^2 V(\bar{y}_k). \tag{5}$$

To determine the optimized stratum size and the sample size in each stratum, we tried to maximize the total sample size while ensuring that the samples are not spatially clustered by using the Clark–Evans aggregation index[74]. The optimized solution is to select 1 sample from each 1° × 1° stratum to maintain the spatial randomness, which results in around 600 samples in each year (Supplementary Figs. 4 and 5) except in 2008 (when we have only ~400 samples). However, the variance of each stratum cannot be estimated from Eq. (5) as we have only one sample in each stratum. We relied on the bootstrapping method by creating the random subsets repeatedly from the original GLAS data to form a distribution of the regional mean. Another concern for implementing this approach and the validity of the estimates is the missing data for some strata, particularly in 2008. Assuming that vegetation changed little in these strata with no valid GLAS observations, we used a gap-filling method by simply taking the estimations from the previous year for strata with missing data. This approach allows statistically valid estimates but provides a conservative estimate of change caused by the gap filling.

**Stratified ordinary kriging**. The design-based SRS method estimates the variance by treating the sampling procedure as independent events. But in the model-based perspective, the observations from spatial samples are correlated because of the nature of spatial autocorrelation[73]. By considering the covariance between samples, we can estimate the spatial mean and variance of region $k$ as

$$\bar{y}_k = \Sigma_{i=1}^{n} \lambda_i \bar{y}_i, \tag{6}$$

$$V(\bar{y}_k) = \Sigma_{i=1}^{n} \lambda_i^2 \sigma_i^2 + 2\Sigma_{i<j} \lambda_i \lambda_j C_{i,j}, \tag{7}$$

where $\lambda_i$ is the kriging weight associated with sample $i$, $\sigma_i^2$ is the sample variance, and $C_{i,j}$ is the covariance between samples. In the variogram-based ordinary kriging models, the covariance is a measure dependent only on sampling distance,

$$C[y(\mathbf{x_i}), y(\mathbf{x_j})] = C(h) = \sigma^2 - \gamma(h), \tag{8}$$

where $h = ||\mathbf{x_i} - \mathbf{x_j}||$ is the distance between $\mathbf{X_i}$ and $\mathbf{X_j}$, and $\gamma(h)$ is the variogram model with parameters such as range, sill and nugget[75,76]. We estimated the mean and variance from all the original GLAS samples using ordinary kriging in each 1° × 1° stratum, and estimated the global mean/variance using Eqs. (4) and (5) from each stratum. For missing data, we applied the same gap-filling method used in the SRS method.

**Spatial modeling with filled data gaps**. Using medium-resolution satellite products from MODIS and QuikSCAT, we extrapolated the GLAS Lidar sampling to a spatially continuous map at 5-km resolution annually from 2003 to 2008. MAIAC EVI represents information about the canopy structure of tropical forests and their potential state of disturbance[27]. QSCAT backscatter contains mixed information of canopy structure and water status. From the temporal mean and variation of these products, we were able to track the annual changes of the forests. With the fixed layers of VCF_max representing the effective canopy cover, plus the annual mean and variation from MAIAC EVI and QSCAT backscatter, we mapped the height

metrics over the Amazon forests using the bagged decision trees (random forest) method[77]. The training data were from the mean GLAS RH metrics estimated using the stratified ordinary kriging (SOK) method at 5-km resolution. The predicted variance of spatial modeling with filled data gaps (SMF) can also be estimated from bootstrapping method due to the richness of the input information and the use of ensemble model. We used the quantile regression forests[78] to estimate the variance of prediction as it keeps the estimation distribution at each leaf node. The gap-filling procedure was similar to what we did in the SRS and SOK methods. Knowing the available 5-km training pixels derived from SOK for each year, we first calculated the median number of valid training pixels (denoted as $\bar{N}_m$) across all $1° \times 1°$ strata (s) from 2003 to 2008, and weighted by $VCF_{max}$ for each 1-deg stratum to be the required valid sample in each stratum, $\bar{N}_s = \bar{N}_m \times VCF_{max}$. Second, the calculated $\bar{N}_s$ was fixed in our modified bagging procedure for each stratum and each year. We randomly drew $\bar{N}_s$ samples for each tree model in the ensemble. When the actual number of samples ($n_s$) for any stratum of a particular year is less than the $\bar{N}_s$, the rest ($\bar{N}_s - n_s$) was drawn randomly from the pool of training pixels retrieved from the previous one (for year 2004 and 2005), two (for year 2006 and 2007) or three years (for year 2008). The strategy in 2003 was slightly different as we filled the gaps in 2003 using training pixels in 2004. The gap-filling procedure ensured that the training samples were spatially balanced and not biased towards any specific region in the Amazon.

**Carbon calculation.** Using ground-calibrated Lorey's height of GLAS from the global dataset[22], we rebuilt the GLAS-derived Lorey's height (LH) specific to the Amazon. Using the RH metrics, we found the best loglinear relationship between LH and RH to be

$$LH = 1.520 \times RH30^{-0.036} \times RH90^{0.828}, \quad (9)$$

where RH30 and RH90 are the heights of 30-percentile and 90-percentile energy returns above ground. The choice of predictor variables were from Lasso regression[79] to keep the minimum number of independent features while ensuring the same level of prediction accuracy by performing 10-fold cross validations.

The further use of benchmark AGB map (in 5-km resolution) of tropical forests estimated from a combination of data, including 4079 in situ inventory plots, satellite GLAS samples, and optical/microwave imagery[1], allowed us to build a relationship between GLAS-derived LH and AGB values for each 5-km pixel. The selection of GLAS training pixels followed the same criteria in the SMF method. Using valid 5-km LH values (2004-2007; Supplementary Discussion) calculated from SOK, we built the one-to-one relationship between AGB and LH with an average uncertainty of ~52 Mg ha$^{-1}$:

$$AGB = 28.78 \times (wd \times LH)^{0.81}, \quad (10)$$

where wd is the estimated wood density accounting for the regional differences in the Amazon basin. With these GLAS-derived AGB values in GLAS locations, we estimated the carbon changes using the same sampling methods. The terrestrial carbon density was calculated using the following equation[1]:

$$CD = \frac{AGB + 0.489AGB^{0.89}}{2}. \quad (11)$$

And the total carbon stock was calculated by multiplying the area of tropical forests (TotC = $\iint CD\, dA$). We used these calculations to approximate the carbon changes in the Amazon forests.

It is important to note that the use of the benchmark map[6] or any other maps of calibration functions may change the absolute value of carbon stocks in the Amazon basin slightly but will not significantly impact the changes of carbon stocks and any potential trends in the carbon stock changes.

**Calculation of emission and forest carbon change.** With the help of MODIS burned area product and the GFC loss data from Landsat, we were able to identify pixels with fire or deforestation. From these pixels in each year, we estimated the emission from deforestation and fire annually. Both datasets were upscaled to 5-km spatial resolution by calculating the area fraction of the fire or deforestation in each 5-km pixel (denoted as $A_f$ for fire, and $A_d$ for deforestation). To find the contribution of emission events, we developed three scenarios.

For pixels with only wildfires happening, we calculated the emission $E_f$:

$$E_f = \iint CD \cdot e_f \cdot dA_f. \quad (12)$$

For pixels with only deforestation, we calculated the emission $E_d$:

$$E_d = \iint CD \cdot e_d \cdot dA_d. \quad (13)$$

For pixels with both fire and deforestation, the emission $E_{fd}$ is

$$E_{fd} = \iint CD \cdot (e_f \cdot dA_f + (1 - e_f)dA_d) \quad (14)$$

where $e_f$ and $e_d$ are the emission efficiency (fraction of carbon release to the atmosphere) for fire and deforestation, respectively. We used the numbers from

literature[32,33] but allowed a fairly large variation: $e_f = 0.3 \pm 0.1$ and $e_d = 0.8 \pm 0.1$. Without considering the covariance between emission factors and the retrieved carbon density at 5-km resolution, we obtained a rough estimation of emissions from fire and deforestation for each year.

The map of forests excluding fire and deforestation (Fig. 3a) were further calculated using the threshold of 1% for each pixel, i.e., if a pixel (25 km$^2$) was identified to have at least 1% deforestation or fire activity, it was no longer intact forest and assigned to deforestation or fire category instead. However, our calculation of total emission of fire and deforestation included all pixels of tropical forests, even though the contribution is less than 1% in some of the pixels. The calculations of intact forest carbon (or structure) change, however, used this threshold of 1% to identify intact forests only at the 5-km spatial resolution.

**Spatial pattern of rainfall.** We stratified the Amazon forests into five different regions based on the rainfall pattern of the TRMM product (Fig. 1a). Monthly CWD values were first calculated from the rainfall data[8,14,80],

$$WD_m = WD_{m-1} - E + P_m \quad when\ WD_m < 0, \quad (15)$$

where $WD_m$ is the water deficit of current month (unit: mm month$^{-1}$), and it equals the water deficit from previous month ($WD_{m-1}$) minus the forest evapotranspiration $E$ (approximated as 100 mm month$^{-1}$), and plus the total rainfall of the current month ($P_m$). To delineate the region that was impacted by the 2005 drought, we also calculated the dry-season (JAS) anomaly of 2005 (Supplementary Fig. 6):

$$Anom(2005) = \frac{WD_{JAS}(2005) - Mean(WD_{JAS})}{Std(WD_{JAS})}, \quad (16)$$

where $WD_{JAS}$ is the CWD over July, August, and September for each year, Mean ($WD_{JAS}$) and Std($WD_{JAS}$) are the long-term mean and standard deviation of $WD_{JAS}$ over the period from 2000 to 2009. The CWD anomaly of other years were included for comparison (Supplementary Fig. 7).

We defined the ED region when Anom(2005)<−2, the SD region when −2 ≤ Anom(2005)<−1, the Moderate water Deficit of the south (MD) when Anom (2005)>−1 and Mean($WD_{JAS}$)<−50, the Light water Deficit of the northeast (LD) when Anom(2005)>−1 and −50<Mean($WD_{JAS}$)<0, and the No water Deficit region (ND) when Mean($WD_{JAS}$)>0.

**Uncertainty of regional estimations.** We use three different methods to sample and interpret the GLAS waveforms (Sampling strategy, Supplementary Fig. 8a). Each sampling strategy has its own assumptions. SRS is the most conservative way of estimating mean height with large uncertainty, but unbiased due to the spatially balanced samples; SOK assumes no dependence between blocks and the mean estimation in each block is from ordinary kriging; SMF using QSCAT and MODIS assumes that the GLAS sampling in each 5-km pixel can well represent the dynamics of tree heights, but the prediction uncertainty from geographically remote samples is much larger than the error predicted from nearby samples measured at a different time. We chose to present the results from SMF, as the changes in SMF were the most conservative.

To further strengthen our analysis, we performed an independent check of the relative changes in top canopy height ($H_G$) measured from GLAS sensor

$$H_G = |R_{beg} - R_{ld}| + Elev_G - Elev_S, \quad (17)$$

where $Elev_G$ and $Elev_S$ are the elevations obtained from GLAS and SRTM, respectively, both of which are variables of the GLAH14 product, $R_{beg}$ and $R_{ld}$ are the range offsets of the waveform for signal beginning and land elevation. The retrieved variable $H_G$ represents the elevation of top canopy height relative to the SRTM elevation. Although AGB and carbon numbers cannot be derived from the $H_G$ metric because SRTM elevation is not the true ground, the annual change of $H_G$ is independent of the uncertainty in our ground detection. However, due to geolocation errors and terrain topography, the uncertainty is associated with the defined reference surface elevation (in our case, the SRTM height). Therefore, we performed a further filter and kept data only with reasonable temporal variation (Std($H_G$)<5 meters). Results of SRS and SMF sampling methods (Supplementary Fig. 8b and c) show that the $H_G$ has a continuous decreasing trend since 2004, similar to what we found in the main manuscript.

**Uncertainty associated with GLAS data filtering.** GLAS data filtering is also an important preprocessing step in our study to reduce the uncertainty and avoid drawing biased conclusions from noisy data. The evolution of the filtering steps (Supplementary Fig. 9) shows that the original data without any filtering present an even more drastic downward trend. Our data screening procedure produced a more conservative, yet still significantly negative trend for interannual changes in the Amazon forests.

Among all the filters, seasonal filtering of the GLAS data played an important role and dramatically changed the interannual variations. The GLAS instrument, during its operational period (2003–2008), acquired data mainly in three seasons—(A) Feb–Mar, (B) May–Jun, and (C) Oct–Nov. However, only seasons A and C

have continuous observations throughout the 6 years. Season B has only valid data in 2004, 2005, and 2006.Because of the seasonal effect existing in GLAS data[81], we decided not to use the annual average of all seasons. The rest two seasons (A and C) are also different, especially in 2005, when season A captured the forest before drought, whereas season B got retrievals after drought. The size of GLAS samples in season A is much smaller than the total size in season C, which makes the uncertainty calculated for season A much larger (Supplementary Fig. 10a and 10b). The larger uncertainty in season A is mainly due to the lack of enough observations, and such large uncertainty makes our change detection harder and not reliable. By plotting the total carbon changes of both seasons, A and C together (Supplementary Fig. 10c and 10d), we see that the uncertainty in season C appears to be smaller and hence better for detecting changes in carbon stocks across the Basin and over the period of the study. It is also worth noting that the seasonal phenology changed from a general increase in carbon from A to C (may not be significant) before the drought, to a general decrease after the drought, particularly in the South (Supplementary Fig. 10d). This finding suggests that the drought event may alter the seasonal phenology to some extent. These findings confirmed that the use of season C (Oct–Nov) data was most suitable for our interannual analysis of canopy structure and carbon changes during the 2005 Amazon drought. Consistency and abundance of data acquired in season C ensured the post-drought decline of carbon concluded in this study was not due to any seasonal variation.

**Data availability**. The data for supporting our findings of this study are publicly available at: https://lpdaac.usgs.gov/; http://www.scp.byu.edu/data.html; https://pmm.nasa.gov/ and https://nsidc.org/data/icesat/data.html.

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

## Acknowledgements

The research was partially supported by NASA Terrestrial Ecology grant at the Jet Propulsion Laboratory, California Institute of Technology and partial funding to the UCLA Institute of Environment and Sustainability from previous National Aeronautics and Space Administration and National Science Foundation grants. The authors thank NSIDC, BYU, USGS, and NASA Land Processes Distributed Active Archive Center (LP DAAC) for making their data available.

## Author contributions

Y.Y., S.S.S., and L.X. designed the study, analyzed data, and wrote the paper. Y.Yu and S. C. provided technical support for GLAS data extraction. S.C., N.P., R.K., M.K., Y.K., and R.B.M. contributed with ideas, writing, and discussions.

## Additional information

**Competing interests:** The authors declare no competing interests.

