## [Peer Review File · Nature Communications]

Reviewers' comments:

Reviewer #1 (Remarks to the Author):

This ms. presents the results of a study to assess the effects of the severe 2005 western/southern Amazon drought on the forest C sink, using annual basin-wide LiDAR from 2003-2008 to determine changes in forest canopy height in those years, and then used established models to generate estimates of above-ground biomass and total carbon. As an ecologist, and not a state-of-the-art remote-sensing (LiDAR) specialist, I am not knowledgeable enough to determine whether the methods were all appropriate/reliable as used here, although the cited literature certainly suggests that broad-scale LiDAR use is robust for such purposes, and the presented results make good sense and are consistent with other published research on the effects of the 2005 Amazon drought (which are well-referenced in this paper). The figures and table are clear enough. Overall, this is a rather straightforward study, with credible, important results. However, I do have several general and specific comments about some ways to improve this manuscript.

First of all, the manuscript would benefit from additional editing to improve numerous (although minor) English grammar issues that would help the readability and clarity various places.

Lines 56-61: These two sentences don't quite make sense as written (copied below), and framing of the manuscript could be strengthened here by including some additional considerations (and associated references).

Your text: Land-surface models, including dynamic global vegetation models (DGVMs), may be used to predict the large-scale effects of climate extremes on the carbon stocks and dynamics if initialized with the spatial heterogeneity of forest biomass and productivity over the landscape and include plant physiological responses to climate. Therefore, the large-scale effects of droughts, and their legacy, over forests of Amazonia remain uncertain.

If I understand the intent of these two sentences (and I'm not sure that I do), then here's a possible change to the first sentence that provides better framing for this paper, with associated references:

"The ability of land-surface models, including dynamic global vegetation models (DGVMs), to project the broad-scale effects of climate extremes on Amazon forest C stocks and dynamics requires initialization with data on the spatial heterogeneity of forest biomass and productivity across the landscape, but such models currently are limited by many factors (Joetzer et al. 2014, Allen et al. 2015), including a general lack of realistic tree mortality functions (McDowell et al. 2013) and uncertainties associated with plant physiological responses to CO₂ enrichment (Sitch et al. 2015). Therefore, the large-scale effects of droughts, and their legacy, over forests of Amazonia remain uncertain."

Allen et al. 2015. On underestimation of global vulnerability to tree mortality and forest die-off from hotter drought in the Anthropocene. *Ecosphere* 6(8):129. <http://dx.doi.org/10.1890/ES15-00203.1>.

Joetzer, et al. 2014. Predicting the response of the Amazon rainforest to persistent drought conditions under current and future climates: a major challenge for global land surface models. *Geoscientific Model Development Discussions* 7:5295–5340.

McDowell, N. G., et al. 2013. Evaluating theories of drought-induced vegetation mortality using a multimodel-experiment framework. *New Phytologist* 200: 304–321.

Sitch et al. 2015. Recent trends and drivers of regional sources and sinks of carbon dioxide. *Biogeosciences* 12:653–679.

Lines 112-114: Your sentence: Therefore, the decline of average RH90 in ED and SD regions suggests that on the average, the canopy trees (large trees) have been impacted significantly and beyond the natural steady-state dynamics of forests in each region.

What is the long-term evidence that these short-term drought effects are "beyond the natural steady-state dynamics of forests in each region" ? Droughts are historically part of the natural dynamics of tropical moist forests, so justify or delete this phrase.

This sentence also could benefit from additional contextual information that supports your observations, such as:

Therefore, the decline of average RH90 in ED and SD regions suggests that many large canopy trees have been impacted significantly, consistent with theoretical and empirical evidence that larger trees are most affected by extreme drought events (McDowell and Allen 2015, Bennett et al. 2015).

Bennett et al. 2015. Larger trees suffer most during drought in forests worldwide. *Nature Plants* 1, Article number: 15139. doi:10.1038/nplants.2015.139.

McDowell, N.G., and C.D. Allen. 2015. Darcy's law predicts widespread forest mortality under climate warming. *Nature Climate Change* 5:669-672.

Lines 180-184: Your concluding paragraph starts out well, but the final sentence seems like a weak ending.

Reviewer #2 (Remarks to the Author):

General Comments) This study uses a remote sensing approach to assess Amazon forest responses to extreme drought over larger spatial scales than have been previously presented. This is an interesting and important paper on a topic of interest to tropical ecologists and climate change scientists. The paper needs some editing for English language and grammar but is generally well written and structured. I cannot assess well the competence of the LiDAR and GIS analyses but they seem appropriate to address the study questions. I have a number of minor comments below, I have two major issues:

1) I question whether the paper presents enough of a leap forward to constitute a Nat Comm paper. From a range of papers we were already fairly confident that (1) the 2005 drought and other droughts of this type had a negative impact on Amazon C storage (e.g: Phillips et al. 2009), (2) this impact was relatively long lived (e.g.: Saatchi et al. 2012; Anderegg et al 2015), and (3) large/tall trees were particularly sensitive (da Costa et al. 2010; Phillips et al. 2010). This paper is really valuable in reinforcing these insights and providing a useful large-scale perspective....but I'm not sure that it's enough.

- Phillips et al. (2009). Drought sensitivity of the Amazon rainforest. *Science* 323: 1344–1347.
- da Costa et al. (2010). Effect of 7 yr of experimental drought on vegetation dynamics and biomass storage of an eastern Amazonian rainforest. *New Phytologist* 187: 579–591.
- Phillips et al. (2010), Drought–mortality relationships for tropical forests. *New Phytologist*, 187: 631–646.
- Anderegg et al. (2015). Pervasive drought legacies in forest ecosystems and their implications for carbon cycle models. *Science* 349: 528-532
- Saatchi, et al. (2012). Persistent effects of a severe drought on Amazonian forest canopy. *PNAS* 110: 565-570.

2) The relationship between tree height and forest carbon, which is used to extrapolate temporal shifts in forest carbon before, during and after the 2005 drought appears to be taken from the relationship between height and carbon when comparing different spatial locations in the Amazon forest (according to the supplementary material). Is this correct? I think that it may be a mistake to take this relationship between two variables over space to extrapolate over time. This is because variation in maximum tree height across space is an indicator of a large number of broader ecosystem differences many of which influence C storage, whereas what we know of drought is that it will disproportionately target the tallest trees whilst leaving many of the other ecosystem components relatively intact (or at least changing them in different ways than observed across spatial gradients). For example, western amazon forests generally have lower maximum tree heights than eastern amazon forests but they also generally have more, more dynamic, smaller stemmed individuals all of which tend to reduce the total aboveground C stock. So if we take the spatial relationship between max height and biomass we would conclude that a reduction

in max height is linked to quite a dramatic reduction in ecosystem C storage (which would be a fair conclusion for spatial extrapolations), but by contrast, drought-induced reduction of maximum tree height across the eastern Amazon to a level more representative of the western Amazon would not affect any of these secondary properties so I believe the actual reduction in C storage would be smaller than predicted by the spatial relationship between height and biomass.

Lines 67-68) Not sure I'd agree with this. What do you mean by most direct? More direct than measuring diameter manually then estimating biomass with allometric equations? With those references I think you'd be on safer ground to argue something like that LiDAR is a good option for estimating biomass over large spatial scales.

Line 76) What do you mean by "committed partitions"?

Line 86) What does it mean that there was a "decline of forest...structure" how is that quantified?

Line 103) What is the evidence that increasing temperature had a role to play in the observed patterns?

Lines 114) Could drought itself have altered seasonal phenology, so effectively altering the seasonal stage surveyed from year to year?

Lines 118-119) The sentence doesn't quite make sense..the changes in AGB point to what exactly?

Line 126) "while South declining" is not correct English. There are multiple examples of such small errors throughout the manuscript. Not a big deal really. But still, get this corrected.

Line 166) This is quite vague, what kinds of changes in understorey growth are you referring to, are there any references supporting this idea?

Figure 1b) I don't understand this panel. What is the green color denoting? Why the choice of date ranges for the orange, blue and pink columns, why not individual years and what happened to 2003 and 2008? Why have cumulative fluxes from 2004, the year before the drought, surely you would want to present these fluxes from the drought year onwards?

Figure 2a) It would probably be clearer if you standardized for absolute biomass by presenting these values as a % of the pre-drought mean biomass

Figures 3b & 3c) Seems strange to present some dates individually then some cumulatively, could easily escape readers that some of the differences are simply because the date ranges are different between columns. Why have cumulative fluxes from 2004, the year before the drought, surely you would want to present these fluxes from the drought year onwards?

Reviewer #3 (Remarks to the Author):

This paper uses a LiDAR time series analysis over the Amazon for five years to document a decline in mean tree height in the southern basin, and links this to the 2005 drought.

The GLAS data provide an intriguing sign of forest height loss over a few years. However, these data cannot ultimately assign this change to drought stress - direct human impacts may be the cause - nor reliably quantify biomass losses, due to methodological issues.

The inference of a drought effect is weak – forest degradation (e.g. logging, unobserved by fire and optical monitoring) could be driving the annual height losses through the period. We know that forest degradation is a large but poorly observed term in the carbon balance of the Amazon. Degradation could be linked to climate/drought stress, and forest plot data support this hypothesis. However, the observation window for GLAS is too short to support the key conclusion (and title) of this paper. The decline in the C sink in the southern basin has been linked to drought in this area, but equally plausibly could be linked to the higher fire count and land use change activities that occur in the southern basin.

With respect to the GLAS methodology, it would help to have a concrete validation of height detection – e.g. against airborne LiDAR? I could not see a reference for this.

The LiDAR estimates of biomass have an estimated error of 52 Mg/ha – which is large enough to make change detection highly challenging. Bias is unreported, and may vary in time and space. Independent validation of remotely sensed biomass produced from GLAS against ground plots reveals marked divergences, so bias is likely. These biases may prevent robust estimates of annual changes in biomass in intact forests.

These estimation problems are compounded by gaps in GLAS coverage – particularly in 2008 when data are sparsest. It is confusing that the biomass estimates for 2008 do not register an increase in uncertainty in e.g. fig 2, which one would expect from the reduced information for this year.

I have some concerns about the selection of the timing of height measurement in October, as using a single common date across the Amazon basin risks biasing the analysis further. Calculation of water deficit is a complex output of precipitation, rooting depth, atmospheric demand and plant physiology, which vary in space, that is not captured in the method here as far as I can tell.

Reviewer comments and author responses

Reviewer #1 (Remarks to the Author):

This ms. presents the results of a study to assess the effects of the severe 2005 western/southern Amazon drought on the forest C sink, using annual basin-wide LiDAR from 2003-2008 to determine changes in forest canopy height in those years, and then used established models to generate estimates of above-ground biomass and total carbon. As an ecologist, and not a state-of-the-art remote-sensing (LiDAR) specialist, I am not knowledgeable enough to determine whether the methods were all appropriate/reliable as used here, although the cited literature certainly suggests that broad-scale LiDAR use is robust for such purposes, and the presented results make good sense and are consistent with other published research on the effects of the 2005 Amazon drought (which are well-referenced in this paper). The figures and table are clear enough. Overall, this is a rather straightforward study, with credible, important results. However, I do have several general and specific comments about some ways to improve this manuscript.

Author response: We thank the reviewer for the concise summary of our work, and the acknowledgement of our results regarding the basin-wide LiDAR analysis. We are also grateful of the reviewer's constructive comments, suggestions and improvements done for our paper. following the reviewer's technical suggestions and comments, we have been able to significantly improve the manuscript in the revised version.

First of all, the manuscript would benefit from additional editing to improve numerous (although minor) English grammar issues that would help the readability and clarity various places.

Author response: The manuscript has been modified extensively to improve grammar issues and make the paper more readable.

Lines 56-61: These two sentences don't quite make sense as written (copied below), and framing of the manuscript could be strengthened here by including some additional considerations (and associated references).

Your text: Land-surface models, including dynamic global vegetation models (DGVMs), may be used to predict the large-scale effects of climate extremes on the carbon stocks and dynamics if initialized with the spatial heterogeneity of forest biomass and productivity over the landscape and include plant physiological responses to climate. Therefore, the large-scale effects of droughts, and their legacy, over forests of Amazonia remain uncertain.

If I understand the intent of these two sentences (and I'm not sure that I do), then here's a possible change to the first sentence that provides better framing for this paper, with associated references: "The ability of land-surface models, including dynamic global vegetation models (DGVMs), to project the broad-scale effects of climate extremes on Amazon forest C stocks and dynamics requires initialization with data on the spatial heterogeneity of forest biomass and productivity across the landscape, but such models currently are limited by many factors (Joetzer et al. 2014, Allen et al. 2015), including a general lack of realistic tree mortality functions (McDowell et al. 2013) and uncertainties associated with plant physiological responses to CO2 enrichment (Sitch et al. 2015). Therefore, the large-scale effects of droughts, and their legacy, over forests of Amazonia remain uncertain."

Allen et al. 2015. On underestimation of global vulnerability to tree mortality and forest die-off from hotter drought in the Anthropocene. *Ecosphere* 6(8):129. <http://dx.doi.org/10.1890/>
Joetzjer, et al. 2014. Predicting the response of the Amazon rainforest to persistent drought conditions under current and future climates: a major challenge for global land surface models. *Geoscientific Model Development Discussions* 7:5295–5340.
McDowell, N. G., et al. 2013. Evaluating theories of drought-induced vegetation mortality using a multimodel-experiment framework. *New Phytologist* 200:304–321.
Sitch et al. 2015. Recent trends and drivers of regional sources and sinks of carbon dioxide. *Biogeosciences* 12:653–679.

Author response: We agree with the reviewer that the rewriting of the sentences contains more solid information on the model developments. We greatly appreciate the reviewer’s effort on the language modification and the citations, which strengthens the introduction of our study background.

Lines 112-114: Your sentence: Therefore, the decline of average RH90 in ED and SD regions suggests that on the average, the canopy trees (large trees) have been impacted significantly and beyond the natural steady-state dynamics of forests in each region.

What is the long-term evidence that these short-term drought effects are “beyond the natural steady-state dynamics of forests in each region” ? Droughts are historically part of the natural dynamics of tropical moist forests, so justify or delete this phrase.

This sentence also could benefit from additional contextual information that supports your observations, such as:

Therefore, the decline of average RH90 in ED and SD regions suggests that many large canopy trees have been impacted significantly, consistent with theoretical and empirical evidence that larger trees are most affected by extreme drought events (McDowell and Allen 2015, Bennett et al. 2015).

Bennett et al. 2015. Larger trees suffer most during drought in forests worldwide. *Nature Plants* 1, Article number: 15139. doi:10.1038/nplants.2015.139.

McDowell, N.G., and C.D. Allen. 2015. Darcy’s law predicts widespread forest mortality under climate warming. *Nature Climate Change* 5:669-672.

Author response: We thank the reviewer’s great help for us to find supporting evidence from literature. We agree with the reviewer that the decline of average RH90 indicates the stronger impact on large trees by these extreme drought events.

Lines 180-184: Your concluding paragraph starts out well, but the final sentence seems like a weak ending.

Author response: We have modified the last sentence as follows: “With detailed eco-hydrological studies of forest function from a combination of widespread ground plots and repeated observations from space, the underlying causes of these changes and their spatial extent and long-term effects can be explored with less uncertainty in future.”

Reviewer #2 (Remarks to the Author):

General Comments: This study uses a remote sensing approach to assess Amazon forest responses to extreme drought over larger spatial scales than have been previously presented. This is an interesting and important paper on a topic of interest to tropical ecologists and climate change scientists. The paper needs some editing for English language and grammar but is generally well written and structured. I cannot assess well the competence of the LiDAR and GIS analyses but they seem appropriate to address the study questions. I have a number of minor comments below, I have two major issues:

Author response: We thank the reviewer for the short summary of our work, and appreciate that the reviewer found our work “interesting and important”. The manuscript has been modified extensively to improve grammar issues and make the paper more readable.

1) I question whether the paper presents enough of a leap forward to constitute a Nat Comm paper. From a range of papers we were already fairly confident that (1) the 2005 drought and other droughts of this type had a negative impact on Amazon C storage (e.g: Phillips et al. 2009), (2) this impact was relatively long lived (e.g.: Saatchi et al. 2012; Anderegg et al 2015), and (3) large/tall trees were particularly sensitive (da Costa et al. 2010; Phillips et al. 2010). This paper is really valuable in reinforcing these insights and providing a useful large-scale perspective....but I'm not sure that it's enough.

- Phillips et al. (2009). Drought sensitivity of the Amazon rainforest. *Science* 323: 1344–1347.
- da Costa et al. (2010). Effect of 7 yr of experimental drought on vegetation dynamics and biomass storage of an eastern Amazonian rainforest. *New Phytologist* 187: 579–591.
- Phillips et al. (2010), Drought–mortality relationships for tropical forests. *New Phytologist*, 187: 631–646.
- Anderegg et al. (2015). Pervasive drought legacies in forest ecosystems and their implications for carbon cycle models. *Science* 349: 528-532
- Saatchi, et al. (2012). Persistent effects of a severe drought on Amazonian forest canopy. *PNAS* 110: 565-570.

Author response: As mentioned by the reviewer, there is enough evidence that large trees are being impacted by droughts disproportionately. The impacts are reflected either in defoliation, canopy damage or mortality. The evidences are usually from chronosequence field measurements at the plot scale or remote sensing data sensitive to leaf greenness, water content or structure. The plot data also provide some estimates of biomass loss but the numbers cannot be readily extrapolated over the entire Amazon basin because of the sample size and distribution, and variations of environmental conditions and species composition. Therefore, the drought impacts on carbon loss from ground measurements and extrapolation often have large uncertainty.

So far, the evidence of carbon/biomass decline during and after the 2005 Amazon drought was based mainly on limited field measurements. But, large-scale estimation of carbon loss over the entire Basin requires repeated measurements of a sufficiently large number of samples distributed systematically or statistically across the Basin. Furthermore, remote sensing data, including optical data and high-frequency Radar backscatter measurements, are more sensitive to the characteristics of top canopy level in the tropics, and less to the entire vegetation structure and biomass and cannot

provide changes of forest carbon loss or change directly. Here, we focus on using systematic sampling of space-borne LiDAR data to detect changes of the forest vertical structure across the entire Amazon to predict the basin-wide changes of aboveground carbon density due to the 2005 drought. We are not aware of any previous studies that have demonstrated the precise large-scale changes of carbon budget over the Amazon basin after the recent droughts.

The relationship between tree height and forest carbon, which is used to extrapolate temporal shifts in forest carbon before, during and after the 2005 drought appears to be taken from the relationship between height and carbon when comparing different spatial locations in the Amazon forest (according to the supplementary material). Is this correct? I think that it may be a mistake to take this relationship between two variables over space to extrapolate over time. This is because variation in maximum tree height across space is an indicator of a large number of broader ecosystem differences many of which influence C storage, whereas what we know of drought is that it will disproportionately target the tallest trees whilst leaving many of the other ecosystem components relatively intact (or at least changing them in different ways than observed across spatial gradients). For example, western amazon forests generally have lower maximum tree heights than eastern amazon forests

but they also generally have more, more dynamic, smaller stemmed individuals all of which tend to reduce the total aboveground C stock. So if we take the spatial relationship between max height and biomass we would conclude that a reduction in max height is linked to quite a dramatic reduction in ecosystem C storage (which would be a fair conclusion for spatial extrapolations), but by contrast, drought-induced reduction of maximum tree height across the eastern amazon to a level more representative of the western Amazon would not affect any of these secondary properties so I believe the actual reduction in C storage would be smaller than predicted by the spatial relationship between height and biomass.

Author response: The reviewer raised a valid concern about the allometric relationship between LiDAR-derived tree height metrics and the corresponding AGB/carbon storage in these forests. In our paper, we addressed this problem in the following aspects:

(1) We did not simply use the max tree height as the proxy for forest carbon. Instead, we used the Lorey's height adopted by (Lefsky 2010), which is the basal-area-weighted height that considers both the horizontal as well as vertical structure of the forest canopy. The use of this metric minimized the biases introduced by height-only metrics in earlier work, and was proved to have more direct correlation with biomass and carbon in various types of forests.

(2) Our model-selection procedure from Lasso regression (Supplementary Information Section 1.2.3) found 2 most significant RH metrics related to Lorey's height among a series of RH metrics (RH10 to RH90). And these two RH (RH30 and RH90) variables represent both the understory (less than 10 meter on average) and overstory (more than 25 meters on average) in the tropical forest canopy. Although the coefficient associated with RH90 is much higher (SI Eq. 9) – suggesting the dominant effect of large trees in estimating forest biomass – the RH30 coefficient is in fact also significant, but with negative impact on the total biomass (the coefficient sign is negative). We can imagine a situation with the fixed RH90 and two different RH30 values. The case with the higher RH30 indicates more canopy elements allocated in upper canopy, thus less understory forest and consequently smaller total AGB, compared to the case of the same RH90

but lower RH30 measurements. The use of 2 metrics instead of 3 and more were also well considered in our regression analysis, as we found statistically insignificant improvements of using more regression variables due to the tight correlations between several RH metrics.

(3) There may be the probability of bias in quantifying the temporal changes from using a spatially-calibrated relationship. Especially, when the available training data have no significant information about the drought-induced structural changes and corresponding C storage change, the biomass model relationship can create a bias in the final estimation of biomass change. In our analysis, we made sure that (a) the original model was developed by including both the old growth forests, as well as the secondary forests of a range of ages and biomass densities (S. S. Saatchi et al. 2011; Lefsky 2010; Lefsky et al. 2007), so that the estimation is not extrapolated from regions with no measurements; (b) we calibrated the data using all available Lorey's height from 2004 to 2007, so that the data have more dynamic range and are not particularly biased towards a certain year's measurements; (c) we assumed that the year-to-year changes are independent, though in reality temporal dependence within the same pixel exists, but the independent assumption helps to produce a conservative (larger) estimation of uncertainty, thus making the findings of significant changes more convincing.

Reference:

- Lefsky, Michael A. 2010. "A Global Forest Canopy Height Map from the Moderate Resolution Imaging Spectroradiometer and the Geoscience Laser Altimeter System." *Geophysical Research Letters* 37 (15): L15401. <https://doi.org/10.1029/2010GL043622>.
- Lefsky, Michael A., Michael Keller, Yong Pang, Plinio B. De Camargo, and Maria O. Hunter. 2007. "Revised Method for Forest Canopy Height Estimation from Geoscience Laser Altimeter System Waveforms." *Journal of Applied Remote Sensing* 1 (1):013537-013537-18. <https://doi.org/10.1117/1.2795724>.
- Saatchi, Sassan S., N. L Harris, S. Brown, M. Lefsky, E. T.A Mitchard, W. Salas, B. R Zutta, W. Buermann, S. L Lewis, and S. Hagen. 2011. "Benchmark Map of Forest Carbon Stocks in Tropical Regions across Three Continents." *Proceedings of the National Academy of Sciences* 108 (24):9899.

Lines 67-68) Not sure I'd agree with this. What do you mean by most direct? More direct than measuring diameter manually then estimating biomass with allometric equations? With those references I think you'd be on safer ground to argue something like that LiDAR is a good option for estimating biomass over large spatial scales.

Author response: We thank the reviewer for pointing out the misuse of the word "most direct". We meant to compare the LiDAR measurements to other satellite data for large-scale studies. Based on the physical characteristics of LiDAR instruments, we could say that the use of LiDAR remote sensing (RS) for forest structure and carbon estimation is the most effective RS method, For dense tropical forests, LiDAR sensors may be the only techniques that provide information on forest vertical and horizontal structure from penetrating into the forest, which is necessary to provide estimates of the total aboveground biomass. We modified the sentence as suggested by the reviewer, and it now reads, "LiDAR samples of vertical structure of forests are recognized as the most effective remote sensing approach to quantify the above ground forest biomass."

Line 76) What do you mean by "committed partitions"?

Author response: We meant to refer to emissions from deforestation, fire and other disturbance as the gross carbon that are immediately released to the atmosphere, even though in most cases the

carbon is released after few years. We replaced “partitions” by “emissions” to related our terminology similar to others in the literature.

Line 86) What does it mean that there was a “decline of forest...structure” how is that quantified?

Author response: Here we use LiDAR height metrics to represent forest structure. By using GLAS-derived RH90 as a height metric to monitor forest canopy disturbance, we report any disturbance of forest structure by the decline in forest height metrics. We modified the sentence in the text to clarify our point, “We observed a widespread decline of forest height..., according to the forest height metric, RH90 (90-percentile of return energy), derived from GLAS LiDAR waveforms at the end of dry season (Oct-Nov) for each year.”

Line 103) What is the evidence that increasing temperature had a role to play in the observed patterns?

Author response: The statement is based on the published article (Toomey et al., 2011) showing that the MODIS land surface temperature (LST) data detected anomalously high daytime and nighttime canopy temperatures throughout drought-affected regions in both 2005 and 2010 Amazon droughts, which may suggest that the heat stress also played an important role in the two droughts. We have also done a similar analysis (unpublished) using the newly processed Version-6 MODIS LST, and found (1) exactly the same anomalies as the previous study during the drought years, and (2) a long-term increasing trend of LST in the Amazon region, which could possibly explain the slow recovery of average canopy height that we discovered here using GLAS data. However, since the focus of this paper is the observations from GLAS data, we did not expand the findings of temperature observations, and the cited article did give us a hint of the temperature impact on the Amazon forest biomass declines. We modified the statement in the manuscript to clarify this point much better.

Reference:

Toomey, M., Roberts, D.A., Still, C., Goulden, M.L., McFadden, J.P., 2011. Remotely sensed heat anomalies linked with Amazonian forest biomass declines. *Geophys. Res. Lett.* 38, L19704. <https://doi.org/10.1029/2011GL049041>

Lines 114) Could drought itself have altered seasonal phenology, so effectively altering the seasonal stage surveyed from year to year?

Author response: Most probably drought may not change the overall phenological cycle of trees in the Amazon Basin over a long period. However, defoliated trees due to drought stress may respond to next year rainfall with a delay and may in turn have an impact on the GLAS detected height change depending on the season of data acquisition. To examine the point raised by the reviewer and test the validity of our assumption for using data from the same season, we investigated the effect of seasonality on GLAS data. (SI Fig. 10, SI text Section 2.3).

(1) The GLAS instrument, during its operational period (2003-2008), acquired data mainly in 3 seasons – A. Feb-Mar; B. May-Jun; and C. Oct-Nov. However, only seasons A and C have continuous observations throughout the 6 years. Season B has only valid data in 2004, 2005 and 2006. And because of the seasonal effect existing in the GLAS data (Tang and Dubayah, 2017),

we decided not to use the annual average of all seasons (which may create anomalous patterns in 2004, 2005 and 2006 due to the inclusion of dry-season observations).

(2) The rest two seasons (A and C) are also different, especially in 2005, when season A captured the forest before drought, whereas season B got retrievals after drought. The size of GLAS samples in season A is much smaller than the total size in season C, which makes the uncertainty calculated for season A much larger (SI Fig. 10a and b). The larger uncertainty in season A is mainly due to the lack of enough observations, and such large uncertainty makes our change detection harder and not reliable. However, if we simply take a look at the mean change of total carbon without considering uncertainty, we observe the total carbon decline starting from 2006. This general trend makes sense, as the drought in 2005 happened during the dry season, mainly after the time period of season A.

(3) By plotting the total carbon changes of both seasons A and C together, we can readily conclude that regardless of the season, there is a general trend in total carbon in the Amazon after the 2005 drought. However, the uncertainty in season C appears to be smaller and hence better for detecting changes in carbon stocks across the Basin and over the period of the study. It is also worth noting that the seasonal phenology changed from a general increase in carbon from A to C (may not be significant) before the drought, to a general decrease after the drought, particularly in the South (SI Fig. 10d). This finding suggests, as in the reviewer's comments, that the drought event may alter the seasonal phenology to some extent in a short term.

The above findings have confirmed that the use of season C (Oct-Nov) data is most suitable for our inter-annual analysis of canopy structure and carbon changes impacted by the drought event. Consistency and abundance of data acquired in season C ensures that the post-drought decline of carbon detected in this study was not due to any seasonal variation.

Reference:

Tang, H., Dubayah, R., 2017. Light-driven growth in Amazon evergreen forests explained by seasonal variations of vertical canopy structure. PNAS 114, 2640–2644.
<https://doi.org/10.1073/pnas.1616943114>

Lines 118-119) The sentence doesn't quite make sense..the changes in AGB point to what exactly?

Author response: We have rewritten the sentence: The regional AGB changes in the western and southern Amazon showed significant losses of biomass after the 2005 drought.

Line 126) “while South declining” is not correct English. There are multiple examples of such small errors throughout the manuscript. Not a big deal really. But still, get this corrected.

Author response: We have modified the sentence as: Forests in the Northern Amazon remained relatively unchanged on average, but Southern Amazon forests declined after the drought event.

Line 166) This is quite vague, what kinds of changes in understorey growth are you referring to, are there any references supporting this idea?

Author response: I think our original sentence may be confusing and may not accurately reflect the current calculation of our forest biomass and carbon. Our carbon calculation indeed puts more weight on the tall trees (reflected in the values of RH90), instead of smaller trees (e.g. shown in RH30). The coefficient associated with the understory is often much smaller compared to the one for tall trees (e.g. SI Eq. 9). However, we also found the non-negligible coefficient for small trees (RH30), which added an adverse effect to the biomass/carbon estimation compared to RH90.

Like we discussed in one of the above responses. We can imagine the situation of a fixed RH90, and lower RH30, which means more canopy elements allocated in smaller trees, thus more secondary forest growth, and therefore more total AGB under the circumstance of the same RH90. Of course in our case, both RH90 and RH30 were declining during our observational period, and this phenomenon cannot be easily concluded.

We thus have made an additional SI figure (SI Fig. 2) showing the fraction of GLAS waveform return above or below a certain fixed height. Results show that the fraction of large trees (approximated as % of waveform return above 25 meters) declined during and after 2005 for the entire Amazon basin, similar to the basin-wide total carbon change we showed in the main manuscript (Fig. 2d). However, such a decrease was accompanied by an increase of % waveform return below 10 meters, meaning most of the decline in tall trees was partly compensated by understory growth within 10 meters; a process representing the faster regeneration of understory trees after the disturbance of loss of canopy trees. We modified the manuscript to explain the changes of the structure of tallest part of the forest compared to understory.

Figure 1b) I don't understand this panel. What is the green color denoting? Why the choice of date ranges for the orange, blue and pink columns, why not individual years and what happened to 2003 and 2008? Why have cumulative fluxes from 2004, the year before the drought, surely you would want to present these fluxes from the drought year onwards?

Author response: We have redone our calculations and updated all figures to include data from 2003 to 2008 and reported uncertainty associated with any changes of carbon from these dates.

We kept the cumulative fluxes from 2004, for the purpose of showing more clearly the significant changes for each region, similar to what we did in Fig. 3. Otherwise, the regional changes are more or less similar to the AGB figures in Fig. 2a, as AGB in this study was mainly derived from RH90.

Figure 2a) It would probably be clearer if you standardized for absolute biomass by presenting these values as a % of the pre-drought mean biomass

Author response: We think that the absolute changes of AGB are also important in some circumstances, and therefore have kept the current form of Fig. 2a. However, the relative change of AGB is indeed a good source of information. We have thus added another column in Table 1 to show the relative changes of height, AGB and carbon to the observations of year 2004.

Figures 3b & 3c) Seems strange to present some dates individually then some cumulatively, could easily escape readers that some of the differences are simply because the date ranges are different

between columns. Why have cumulative fluxes from 2004, the year before the drought, surely you would want to present these fluxes from the drought year onwards?

Author response: We thank the reviewer for pointing out our poorly presented figure. In this figure, we wanted to demonstrate the significant changes in the “South”, in contrast to the insignificant changes found in the “North”. If we simply used the year-to-year change, we would not be able to see the significant changes after 2 years onwards. As we also noted in the text, “the uncertainty of this estimate precludes evaluation of biomass decline immediately after the drought event”, but the lagged effect and the prolonged impact of the drought enables us to find a statistically significant estimate of biomass loss one year after the drought. Therefore, we chose to use the cumulative fluxes in this figure and make this phenomenon stand out.

However, in the revised version, we have also added 2 panels showing year-to-year variations and modified the text accordingly, to make the presentation of the cumulative figure more naturally.

Reviewer #3 (Remarks to the Author):

This paper uses a LiDAR time series analysis over the Amazon for five years to document a decline in mean tree height in the southern basin, and links this to the 2005 drought. The GLAS data provide an intriguing sign of forest height loss over a few years. However, these data cannot ultimately assign this change to drought stress - direct human impacts may be the cause - nor reliably quantify biomass losses, due to methodological issues.

Author response: We thank the reviewer for the concise summary of our study, and pointing out some concerns regarding the cause of the drought impact on Amazon forests and our methodology. However, we think our methodology demonstrates the biomass change and uncertainty associated with it and provide ample evidence that the decline of the biomass in the old growth forests is mainly due to the direct impact of drought and not other factors such as deforestation and fire. We included some extra material in the discussion about all possible human-induced changes of forest structure during and after the drought year. Our analysis was organized in 3 stages.

At the first stage, we simply looked at the changes of height directly retrieved from GLAS LiDAR waveforms, without considering any natural cause or human impact. And we found significant declines of tree height (Fig. 1b) in the drought regions (ED, SD and MD). This result is only from satellite observations without calculating biomass, or any attribution of causes. It is also interesting to see that the region ED (extreme drought) had the most significant drop in tree height one and two years after the drought event, while fire, deforestation, degradation (logging, understory fires) events mostly happened in the region MD (comparing Fig. 1a and 3a).

At the second stage, we translated height changes into biomass and carbon. It is true that the use of various allometric equations could generate different results. But in general biomass/carbon values are tightly correlated with dominant tree heights within canopy, and in our case, related more to the RH90 metric. As expected, we found similar patterns of AGB/carbon in each climatic region (Fig. 2a), and combining the regions, we got a nice separation between north and south in the Amazon (Fig. 2b and 2c). It should be noted that the definitions of North and South were calculated from rainfall anomalies (from TRMM data) and did not contain any information of human activities. Deforestation and fire pixels were removed from the analysis.

We attempted to separate the contributions of carbon stock changes at the final stage. We therefore used MODIS Fire product and Landsat-based forest loss, and tried to exclude regions with observable Fire and Deforestation (Fig. 3a). With the threshold set to 1% of each large pixel area (5 km x 5 km), which may have an impact of fire or deforestation, we applied a strict filter to locate pixels within intact forests, potentially removing all forests that may be impacted by biomass loss from understory fire and edge effects from our analysis of biomass changes in intact forests. It may be true that the residual effect from human impact (such as forest degradation not detectable from space) still exists. However, there is no evidence in the literature and from existing data on degradation from logging that was increased or changed after the drought year. Post-degradation understory and forest edge fire, however, may have damaged the forest canopy and therefore caused for the decline of the forest biomass. If these fires are not captured in MODIS hot pixel data, there may be a residual effect of this in the final result. Nevertheless, we believe that this residual effect is very small because all pixels (25 km²) with > 1% deforestation or fire fraction

has been excluded from the analysis. Additionally, we see this effect in the ED region where there is no large scale logging activities. We performed these robust filters to the best of our knowledge.

In addition, based on our reported numbers in Fig. 3b, the drought-affected carbon loss is comparable to, or even larger than, the emission from deforestation. But the latest study (Pearson et al., 2017) shows that degradation effect may contribute to a quarter of the combined emission from deforestation and degradation. Even though there might be residual human impact in our estimation, the observed drought-related carbon loss is much larger than 1.25 times of deforestation emission. We therefore assigned this change to drought stress. Furthermore, degradation will have both carbon loss and immediate carbon gain from regeneration, making the net effect of degradation much smaller. In addition, forest degradation from logging does not necessarily have a downward trend in carbon change, because there is no indication that after 2005 drought, there were increasing areas of forest logging or other human disturbance not accounted in our study.

Reference:

Pearson, T.R.H., Brown, S., Murray, L., Sidman, G., 2017. Greenhouse gas emissions from tropical forest degradation: an underestimated source. *Carbon Balance and Management* 12, 3.
<https://doi.org/10.1186/s13021-017-0072-2>

The inference of a drought effect is weak – forest degradation (e.g. logging, unobserved by fire and optical monitoring) could be driving the annual height losses through the period. We know that forest degradation is a large but poorly observed term in the carbon balance of the Amazon. Degradation could be linked to climate/drought stress, and forest plot data support this hypothesis. However, the observation window for GLAS is too short to support the key conclusion (and title) of this paper. The decline in the C sink in the southern basin has been linked to drought in this area, but equally plausibly could be linked to the higher fire count and land use change activities that occur in the southern basin.

Author response: We agree with the reviewer that some of the effects of forest degradation are not observable from current satellite sensors. As we pointed out in the above response, the contribution of unobservable degradation emission can be < 25% of total emission from deforestation. Based on our calculation of deforestation emission using high-resolution Landsat product (Hansen et al., 2013), and a high emission factor for deforestation, we are confident that most deforestation events were accounted for. However, the carbon loss in the southern basin, excluding fire and deforestation, still showed a significant number – much larger than the possible missing emission from forest degradation.

Reference:

Hansen, M.C., Potapov, P.V., Moore, R., Hancher, M., Turubanova, S.A., Tyukavina, A., Thau, D., Stehman, S.V., Goetz, S.J., Loveland, T.R., Kommareddy, A., Egorov, A., Chini, L., Justice, C.O., Townshend, J.R.G., 2013. High-Resolution Global Maps of 21st-Century Forest Cover Change. *Science* 342, 850–853. <https://doi.org/10.1126/science.1244693>

With respect to the GLAS methodology, it would help to have a concrete validation of height detection – e.g. against airborne LiDAR? I could not see a reference for this.

Author response: As suggested by the reviewer, we have added references showing (1) the consistency of Lidar measurements between GLAS LiDAR and airborne small-footprint LiDAR at well-studied sites comparing a series of LiDAR metrics and derived AGB values (Popescu et al., 2011); (2) the validation of GLAS LiDAR against the airborne waveform (LVIS) LiDAR, small-footprint LiDAR, and field data (Lee et al., 2011); and (3) the effect of various GLAS data filters and the comparison with airborne LiDAR as well as global vegetation height product used in climate models (Los et al., 2012).

Reference:

- Popescu, S.C., Zhao, K., Neuenschwander, A., Lin, C., 2011. Satellite lidar vs. small footprint airborne lidar: Comparing the accuracy of aboveground biomass estimates and forest structure metrics at footprint level. *Remote Sensing of Environment* 115, 2786–2797. <https://doi.org/10.1016/j.rse.2011.01.026>
- Lee, S., Ni-Meister, W., Yang, W., Chen, Q., 2011. Physically based vertical vegetation structure retrieval from ICESat data: Validation using LVIS in White Mountain National Forest, New Hampshire, USA. *Remote Sensing of Environment* 115, 2776–2785. <https://doi.org/10.1016/j.rse.2010.08.026>
- Los, S.O., Rosette, J.A.B., Kljun, N., North, P.R.J., Chasmer, L., Suárez, J.C., Hopkinson, C., Hill, R.A., van Gorsel, E., Mahoney, C., Berni, J.A.J., 2012. Vegetation height and cover fraction between 60° S and 60° N from ICESat GLAS data. *Geosci. Model Dev.* 5, 413–432. <https://doi.org/10.5194/gmd-5-413-2012>

The LiDAR estimates of biomass have an estimated error of 52 Mg/ha – which is large enough to make change detection highly challenging. Bias is unreported, and may vary in time and space. Independent validation of remotely sensed biomass produced from GLAS against ground plots reveals marked divergences, so bias is likely. These biases may prevent robust estimates of annual changes in biomass in intact forests.

Author response: The reviewer may have misunderstood the uncertainty calculation in our study.

(1) The average error of 52 Mg/ha reported in AGB allometry is the pixel-level uncertainty at 5-km spatial resolution. It is indeed large, and therefore, we did not report any significant changes at the pixel level. Instead, we reported regional differences, where in each region, the total number of pixels can be tens of thousands. If all errors are independent, the mean AGB should have an error reduced by \sqrt{N} , where N is the total number of pixels. Therefore, the uncertainty of regional estimates can be hundreds of times smaller.

(2) The existence of spatial autocorrelation could make the error estimate of regional mean larger. We therefore considered the possible spatial autocorrelation in the spatial mapping model using our modified bagging procedure and incorporated the uncertainty estimates in the quantile regression forests (the last paragraph of SI section 1.2.2). The reported uncertainty in our study are thus much larger than the results assuming data independency.

(3) Uncertainty due to random errors associated with the estimates of biomass at the pixel level can become negligible when aggregated over large regions. However, just like the reviewer

mentioned, any systematic bias at the pixel level will accumulate rapidly and can introduce large uncertainty at the regional scales. There are also reports showing systematic bias of GLAS in tree height estimation. To avoid these systematic biases, we therefore calibrated the maps of AGB and carbon using GLAS for each year. It is why we only looked at the changes from 2003 to 2008, whereas the satellite data used for predictions can span over more than a decade.

(4) To avoid possible sampling biases due to missing data of GLAS observations in certain years (the most severe example is in 2008), we also performed the gap-filling procedure based on reasonable assumptions. Without these assumptions and the data gap-filling procedure, our reported numbers may not be robust enough. But the current procedure ensured that at least during the GLAS observational period (2003 – 2008), the reported numbers are consistent with its own calibration system.

These estimation problems are compounded by gaps in GLAS coverage – particularly in 2008 when data are sparsest. It is confusing that the biomass estimates for 2008 do not register an increase in uncertainty in e.g. fig 2, which one would expect from the reduced information for this year.

Author response: We did not see an increase in uncertainty for 2008 (Fig. 2) was because

(1) We performed gap-filling for each year to eliminate sampling biases. Therefore, the sample size after gap-filling is similar for each year and spatially balanced. The procedure of gap-filling is mostly described in SI section 1.2.2. The gap-filling basically assumes no change for “gaps” with missing data. Therefore, if a region has no data observed from the current year, it uses observations from previous years assuming no forest structure change for this particular location. This is the most conservative assumption, as the objective of our study is to detect changes.

(2) To answer the speculation of the reviewer, we did another version of Fig. 2d showing the total carbon changes without using gap-filling (Fig. R1b). The result shows a larger uncertainty for 2008, exactly as the reviewer expected and the larger uncertainty is indeed because of reduced information. However, we can see that the overall uncertainty for each year is in fact much less than our original figure (Fig. R1a). And the magnitude of change (~3 Pg from 2004 to 2008) is much larger than the original estimation (~1 Pg from 2004 to 2008), in another way, proving that our gap-filled method has more conservative estimations.

(3) The reason that we did not use the version in Fig. R1b was to avoid biases introduced by sampling. As we can see from Fig. R1c and SI Fig. 3, the valid samples change from year to year. Without using spatially balanced samples and consistent sample size, we could introduce biases by extrapolating data out of the range of our training data. Furthermore, since the cross-validation of regression model was against the same data set, the ignorance of the missing data would miss the part of uncertainty introduced by that part of the data, which is why the uncertainty in Fig. R1b is smaller than Fig. R1a. On the other hand, the use of gap-filled data with similar sample size brings the same level of variation into each year, and thus the uncertainty (Fig. R1a) appears to be similar.

(4) Of course, in each year, the fraction of gap-filled data is different. We have much more gap-filling in 2008. If we introduce a random error into the gap-filled data, i.e., without assuming that the tree height has no change – instead, changes randomly, we can see that the uncertainty in 2008 is relatively larger (Fig. R1d) – because of more fraction of gap-filling in 2008. However, such assumption (random changes of forest structure) makes the whole case useless, as no significant changes can be detected anyway.

Figure R1. Experiments of different assumptions in finding carbon stock changes of the Amazon Basin. (a) The original figure in main manuscript (Fig. 2d); (b) The same setting without gap-filling in spatial mapping; (c) The histogram comparing the valid data in 2004 and 2008; (d) The same test as panels (a) and (b), but adding more variations (Gaussian noise with 0 mean and 3 meters in σ) to the gap-filled data.

I have some concerns about the selection of the timing of height measurement in October, as using a single common date across the Amazon basin risks biasing the analysis further.

Author response: We understand the reviewer's concern about using data from a single season (Oct/Nov). The choice of a single season in our study was to avoid possible seasonality interfering the signal of inter-annual changes. To make this problem clearer, we have done additional analyses (SI Fig. 10, SI text Section 2.3) to investigate the seasonal effect of the GLAS data.

(1) The GLAS instrument, during its operational period (2003-2008), acquired data mainly in 3 seasons – A. Feb-Mar; B. May-Jun; and C. Oct-Nov. However, only seasons A and C have continuous observations throughout the 6 years. Season B has only valid data in 2004, 2005 and 2006. And because of the seasonal effect existing in the GLAS data (Tang and Dubayah, 2017), we decided not to use the annual average of all seasons (which may create anomalous patterns in 2004, 2005 and 2006 due to the inclusion of dry-season observations).

(2) The rest two seasons (A and C) are also different, especially in 2005, when season A captured the forest before drought, whereas season B got retrievals after drought. The size of GLAS samples in season A is much smaller than the total size in season C, which makes the uncertainty calculated for season A much larger (SI Fig. 10a and b). The larger uncertainty in season A is mainly due to the lack of enough observations, and such large uncertainty makes our change detection harder and not reliable. However, if we simply take a look at the mean change of total carbon without considering uncertainty, we can see generally total carbon did not decrease until 2006. This general trend makes sense, as the drought in 2005 happened in the dry season of Amazon, usually after the time period of season A.

(3) When we plot the total carbon changes of both seasons A and C together, we see clearly the smaller uncertainty in season C, and the generally downward trend of total carbon in Amazon south after the drought. It is also worth noting that the seasonal phenology changed from a general increase in carbon from A to C (may not be significant) before the drought, to a general decrease after the drought, particularly in the South (SI Fig. 10d). This change of seasonal signal also made us avoid using the combined (2-season) series for the inter-annual change detection.

The above findings have confirmed that the use of season C (Oct-Nov) data is most suitable for our inter-annual analysis of canopy structure and carbon changes impacted by the drought event. The data consistency and abundance in season C have made sure that the post-drought decline of carbon we found in this study was not due to any seasonal variation that could have an additional impact on the change detection.

Reference:

Tang, H., Dubayah, R., 2017. Light-driven growth in Amazon evergreen forests explained by seasonal variations of vertical canopy structure. PNAS 114, 2640–2644.
<https://doi.org/10.1073/pnas.1616943114>

Calculation of water deficit is a complex output of precipitation, rooting depth, atmospheric demand and plant physiology, which vary in space, that is not captured in the method here as far as I can tell.

Author response: We calculated the water deficit from rainfall data derived from the TRMM satellite data. The definition of it was used in several previous studies of tropical forests (Aragão et al., 2007; Saatchi et al., 2013; Feldpausch et al., 2016), and proved to be a well-representative and widely-used quantity for water deficit.

Reference:

Aragão, L.E.O.C., Malhi, Y., Roman-Cuesta, R.M., Saatchi, S., Anderson, L.O., Shimabukuro, Y.E., 2007. Spatial patterns and fire response of recent Amazonian droughts. Geophys. Res. Lett. 34, L07701. <https://doi.org/10.1029/2006GL028946>

Saatchi, S., Asefi-Najafabady, S., Malhi, Y., Aragão, L.E.O.C., Anderson, L.O., Myneni, R.B., Nemani, R., 2013. Persistent effects of a severe drought on Amazonian forest canopy. PNAS 110, 565–570. <https://doi.org/10.1073/pnas.1204651110>

Feldpausch, T.R., Phillips, O.L., Brienen, R.J.W., Gloor, E., Lloyd, J., Lopez-Gonzalez, G., Monteagudo-Mendoza, A., Malhi, Y., Alarcón, A., Álvarez Dávila, E., Alvarez-Loayza, P., Andrade, A., Aragao, L.E.O.C., Arroyo, L., Aymard C., G.A., Baker, T.R., Baraloto, C., Barroso, J., Bonal, D., Castro, W., Chama, V., Chave, J., Domingues, T.F., Fauset, S., Groot, N., Honorio Coronado, E., Laurance, S., Laurance, W.F., Lewis, S.L., Licona, J.C., Marimon, B.S., Marimon-Junior, B.H., Mendoza Bautista, C., Neill, D.A., Oliveira, E.A., Oliveira dos Santos, C., Pallqui Camacho, N.C., Pardo-Molina, G., Prieto, A., Quesada, C.A., Ramírez, F., Ramírez-Angulo, H., Réjou-Méchain, M., Rudas, A., Saiz, G., Salomão, R.P., Silva-Espejo, J.E., Silveira, M., ter Steege, H., Stropp, J., Terborgh, J., Thomas-Caesar, R., van der Heijden, G.M.F., Vásquez Martinez, R., Vilanova, E., Vos, V.A., 2016. Amazon forest response to repeated droughts. *Global Biogeochem. Cycles* 30, 2015GB005133. <https://doi.org/10.1002/2015GB005133>

Reviewers' comments:

Reviewer #1 (Remarks to the Author):

I am satisfied with the author responses to my initial comments.

However, I upon re-reading thru the ms. and associated figures, I have several new comments and questions that I suggest be addressed by the authors.

Comparing the Fig 1-a map vs. Fig 3-a map, note that the substantial areas identified as "Deforested" and "Forest Fire" in Fig 3-a are shown as forest in Fig 1-a, and thus apparently included (?) in your analyses of drought-affected pixels; so, were these land-use-disturbed pixels included in Lidar analyses, or not? Actually, since these land-use-disturbed pixels (by logging and fire) were independently identified, it seems like it would have made sense to have removed them upfront from the Lidar analyses of forest change (which also could help address some of Reviewer #3's concerns) – it also would have been of interest to see what the Lidar analyses of the separated out "Deforested" and "Forest Fire" pixels might indicate about changes in forest structure from those land uses.

Related, in the Fig 1-a legend text, "Non-tropical forests in panel (a) were colored in gray", perhaps for both clarity and precision that sentence should be modified to read "Non-forests as of 2003 were colored grey in panel (a)", since many of the non-grey forest pixels of Fig 1-a are shown to be "Deforested" over the 2003-2008 time period that you assessed.

Re: Fig 1-a, the mapped spatial distribution of drought severity classes is a bit surprising in that there are two directionally-contrasting spatial gradients of severity (in the North, a N-to-S gradient of increasing drought, vs. in the South a S-to-N gradient increasing drought) that seem to clash along an East-West line where ED & SD pixel-patches (reflecting the South gradient) abruptly meet LD or even ND patches (from the North gradient); what is surprising is the complete lack of gradational pixels (particularly MD) across this abruptly contrasting contact zone. I recognize that these are relatively coarse-resolution pixels (so perhaps the "missing" between-class changes occur at within-pixel spatial scales), but nonetheless this striking spatial pattern is at least an interesting pattern that merits an explanation (whether it's climatological, geographical, or methodological).

Re: a comparison of Fig 1-b and Fig 2-a, the relatively lowest stature of MD pixels (e.g., 10% shorter than SD) makes sense, given the transition to drier savanna and deforestation/fire disturbances along this southernmost zone. However, one would expect this lowest-stature canopy in MD areas to translate to the lowest level of AGB, but Fig 2-a indicates otherwise, with SD shown as the lowest AGB zone. Please explain how this can be the case.

Re: Fig 3, to me it would make more sense in panels 3-b/c/d to reverse the +/- direction of the graphed bars, because I don't see how loss of forest C from whatever process can be graphed as a positive "Forest Carbon Change". I suppose you conceived of it as positive "emissions", but that doesn't match or make sense given the current y-axis label.

Also, comparing Fig. 3-c and 3-d "uncertainty" values, I'm wondering if the 04-06 "South Intact" uncertainty value in Fig 3-d should be changed from "+-0.19" to "+- 0.21" (to match the 04-06 value in Fig 3-c, since the rest of these values match between these figures).

Reviewer #2 (Remarks to the Author):

The authors have adequately addressed all of my comments wherever possible. I agree with the authors that their analysis presents the best current estimate of Amazon-wide carbon responses to recent drought events, but it is still true that most of their major findings have been fairly well

tested in several previous studies. This analysis definitely deserves to come out somewhere, i just question whether the greater quality/accuracy/resolution of the dataset in itself merits publication in Nat Comm, given than the actual results are fairly unsurprising given published works.

Reviewer #3 (Remarks to the Author):

The authors have made considerable improvements to the ms, and it is now close to publishable quality. I have some comments that need to be addressed prior to this:

"The decline of carbon stocks started from the hydraulic stress...". This is an inference, because hydraulic stress is not directly measured. It would be better to add "We infer that..." at the start of this sentence.

The text notes that inventory plots suggest the 2005 drought converted forests from a "sink of about 0.71 MgC ha⁻¹ yr⁻¹ to a net source of carbon to the atmosphere of about 1.5 MgC ha⁻¹ yr⁻¹". I suggest that the units are converted to MgC ha⁻¹, to match the numbers from the introductory sentence, and because of the extrapolation challenges of these plot level data noted in the next sentence.

Reviewer comments and author response

Reviewer #1 (Remarks to the Author):

I am satisfied with the author responses to my initial comments. However, I upon re-reading thru the ms. and associated figures, I have several new comments and questions that I suggest be addressed by the authors.

Author response: We are glad that the reviewer is satisfied with our responses, and grateful to the reviewer for giving us valuable comments and suggestions to improve our manuscript. We have revised the manuscript accordingly and have incorporated the reviewer's suggestions.

Comparing the Fig 1-a map vs. Fig 3-a map, note that the substantial areas identified as "Deforested" and "Forest Fire" in Fig 3-a are shown as forest in Fig 1-a, and thus apparently included (?) in your analyses of drought-affected pixels; so, were these land-use-disturbed pixels included in Lidar analyses, or not? Actually, since these land-use-disturbed pixels (by logging and fire) were independently identified, it seems like it would have made sense to have removed them upfront from the Lidar analyses of forest change (which also could help address some of Reviewer #3's concerns) – it also would have been of interest to see what the Lidar analyses of the separated out "Deforested" and "Forest Fire" pixels might indicate about changes in forest structure from those land uses.

Author response: We thank the reviewer for raising this concern about regions of "Deforested" and "Forest Fire". We removed the fire and deforested pixels throughout our 5km analysis of changes of forest structure and carbon. However, we would like to clarifying few points about the figures.

1. Fig. 1a is the drought classification map derived purely from rainfall data, which is the product from TRMM data. The TRMM product has a spatial resolution of 0.25x0.25 degree. At this coarse resolution, we could only set pixels as valid when the majority area within the pixel is covered by tropical forests. If including localized (sub-pixel) activities, such as deforestation and fire events, we would make a highly fragmented map that is unable to represent the entire tropical forests. The main purpose of the figure is to show the footprint of the 2005 drought over the entire Amazon Basin regardless of it land cover type.

2. However, in calculating the forest height change in Fig 1b, we excluded all the pixels that had more than 1% deforestation or fire events. Therefore, the calculations in Fig 1b is consistent with figures 2 and 3 ad hence the regional plots in Fig. 1b and Fig. 2a should correspond to the intact forest pixels as shown in Fig. 3a within each of the 5 regions defined in Fig. 1a. Similarly, the North and South regional plots in Fig. 2b and 2c should follow the regions shown in Fig. 3a as the deforested and fire pixels are excluded. We made sure this information is included in the caption of the figure.

3. In Fig. 3a, we included the "Fire" and "Deforestation" pixels from a finer resolution map to highlight their impacts in calculating the emissions from changes of forest cover and droughts over the Amazon Basin. The fire and deforestation pixels found in Fig. 3a were calculated based

on a finer-resolution map (5x5 km), and set as “Fire” or “Deforestation” once the pixel has more than 1% of these activities detected during the observational period (last paragraph of SI text section 1.2.4). Therefore, using a very conservative threshold (1%), we make sure that all pixels with a small percentage of fire and deforestation are not included in our carbon change analysis of intact forests as a result of droughts. However, this does not mean that the area of land use change and fire are as widespread as the number of pixels removed from the analysis. The “Fire” or “Deforested” pixels may still have up to 99% of intact forests. In Fig. 1a, we intended not to confuse the readers by these sub-pixel effects and therefore selected pixels only if the majority class is the tropical evergreen forest for demonstrating overall areas impacted by droughts.

4. To further clarify the data processing steps, we modified both the main manuscript and SI text to emphasize the use of satellite-based fire and deforestation data before any result of observed forest changes.

5. The reviewer also mentioned the possibility of tracking land-use changes from fire and deforestation pixels. We actually thought about this issue, but in reality, fire and deforestation are sub-pixel events in our 5-km map where the majority belongs to intact forests, and hard to be separated out for change detection. In mixed pixels, we have no reason to attribute all detected carbon loss/gain to a small fraction of land-use changes. Therefore, in Fig. 3, we calculated the fire and deforestation emissions using empirical estimations by introducing emission factors (SI text 1.2.4), instead of directly estimate changes from the map. The emission factor is only applied to the proportion of the pixel identified as deforested or fire from higher resolution imagery.

Related, in the Fig 1-a legend text, “Non-tropical forests in panel (a) were colored in gray”, perhaps for both clarity and precision that sentence should be modified to read “Non-forests as of 2003 were colored grey in panel (a)”, since many of the non-grey forest pixels of Fig 1-a are shown to be “Deforested” over the 2003-2008 time period that you assessed.

Author response: We apologize for the inaccurate use of words in Fig. 1 that caused confusion. This is another similar issue of mixed pixel and spatial resolution. In Fig. 1a, the source of data (TRMM product) is in 0.25x0.25-degree (~25-km) spatial resolution, while the information of deforestation is derived from Landsat (~30-meter), and fire from MODIS (~500-meter). One example could be, in one 25x25-km pixel, only 2% of the pixel experienced fire or deforestation, and the rest 98% region is still intact forests. In Fig. 3a, we would mark it as fire/deforested, as we used a threshold of 1% to eliminate all possible sources that could blur the drought signal. But in Fig. 1a, all colored pixels still represent tropical forests, and it should be correct to say that “Non-tropical forests in panel (a) were colored in gray”. To make the figures less confusing, we added one more sentence in Fig. 3 caption to clarify the use of 1% threshold for the map coloring in Fig. 3a.

Re: Fig 1-a, the mapped spatial distribution of drought severity classes is a bit surprising in that there are two directionally-contrasting spatial gradients of severity (in the North, a N-to-S gradient of increasing drought, vs. in the South a S-to-N gradient increasing drought) that seem to clash along an East-West line where ED & SD pixel-patches (reflecting the South gradient) abruptly meet LD or even ND patches (from the North gradient); what is surprising is the

complete lack of gradational pixels (particularly MD) across this abruptly contrasting contact zone. I recognize that these are relatively coarse-resolution pixels (so perhaps the "missing" between-class changes occur at within-pixel spatial scales), but nonetheless this striking spatial pattern is at least an interesting pattern that merits an explanation (whether it's climatological, geographical, or methodological).

Author response: Regarding the classification of rainfall, we delineated 5 regions using 2 maps: (a) defining ED and SD regions based on the JAS CWD anomaly in 2005; and (b) defining MD, LD and ND mainly from the mean JAS CWD averaged over the based period (2000- 2009) (See SI text 1.2.5).

Therefore, the separation of North and South was mainly due to the use of (b) mean JAS CWD map (see SI Fig. 6b). This map shows most pixels with mean JAS CWD less than 50 were located in the South. Therefore, we can hardly find any MD pixel in the North. This classification scheme provides a categorical thematic map of drought impacted areas and therefore by default have abrupt boundaries.

Furthermore, the changes in the impact of drought in the ED/SD and LD/ND abrupt boundaries is also due to the fact that the impact of droughts are localized to some extent and these local differences are defined by using a threshold on the water deficit derived from TRMM data. Fig. R1, cited the figure from literature (Saatchi et al., 2013), clearly shows the rainfall anomaly from the TRMM analysis with sharp boundaries in western region of Amazonia. This sharp difference is because the anomaly is classified by the number of standard deviation from the mean. The water deficit itself has a more continuous variation over the landscape.

Fig. R1. Comparison of Dry-season Precipitation Anomaly (DPA) and Dry-season Water Deficit Anomaly (DWDA) in the year 2005 (Saatchi et al. 10.1073/pnas.1204651110)

Re: a comparison of Fig 1-b and Fig 2-a, the relatively lowest stature of MD pixels (e.g., 10% shorter than SD) makes sense, given the transition to drier savanna and deforestation/fire disturbances along this southernmost zone. However, one would expect this lowest-stature canopy in MD areas to translate to the lowest level of AGB, but Fig 2-a indicates otherwise, with SD shown as the lowest AGB zone. Please explain how this can be the case.

Author response: We thank the reviewer for raising this concern regarding the inconsistent behaviors of height and AGB in the MD region. Here, the calculation of AGB followed SI Eq. 10, where we also got a correction factor w_d – wood density – for the Lidar heights. Although we found the mean RH90 in the SD region is higher than the MD region, the average wood density is around 0.55, less than the wood density in the MD region that we used (~0.6). Therefore, the resulting AGB in the MD region appeared to be higher instead.

Re: Fig 3, to me it would make more sense in panels 3-b/c/d to reverse the +/- direction of the graphed bars, because I don't see how loss of forest C from whatever process can be graphed as a positive "Forest Carbon Change". I suppose you conceived of it as positive "emissions", but that doesn't match or make sense given the current y-axis label.

Author response: We thank the reviewer for pointing this out. The use of y-axis label was indeed confusing, and therefore we have modified the y-axis label of Fig. 3b, 3c and 3d as "Forest Carbon Emission" to match the signs of bar plots. As the reviewer suggested, the use of "Emission" is more appropriate here, because the main purpose of the figure here is partitioning carbon emissions. We have also modified the figure caption to further explain the meaning of the y-axis direction.

Also, comparing Fig. 3-c and 3-d "uncertainty" values, I'm wondering if the 04-06 "South Intact" uncertainty value in Fig 3-d should be changed from "+-0.19" to "+- 0.21" (to match the 04-06 value in Fig 3-c, since the rest of these values match between these figures).

Author response: First, we have to say that Fig. 3c and 3d are different: Fig. 3c is the annual (year-to-year) contribution of carbon emissions in the South; while Fig. 3d is the cumulative contribution of carbon emissions relative to the year 2004. Therefore, the 2006 change in Fig. 3c in fact means the carbon change from 2005 to 2006, whereas the bars of 04-06 in Fig. 3d indicate the changes from 2004 to 2006. The uncertainty numbers for the rest of the years just happened to be the same for the first 2 significant digits. In fact, if taking forest fire emission as an example, we can see that the uncertainty values between Fig. 3c and 3d are very different. Our sampling strategy and gap filling methods have made the uncertainty estimations in each year the same order of magnitude, but still essentially different.

Reviewer #2 (Remarks to the Author):

The authors have adequately addressed all of my comments wherever possible. I agree with the authors that their analysis presents the best current estimate of Amazon-wide carbon responses to recent drought events, but it is still true that most of their major findings have been fairly well tested in several previous studies. This analysis definitely deserves to come out somewhere, i just question whether the greater quality/accuracy/resolution of the dataset in itself merits publication in Nat Comm, given that the actual results are fairly unsurprising given published works.

Author response: We thank the reviewer for finding our analysis to have the best current estimate of Amazon-wide carbon responses to recent drought events. It is true that the major findings have been tested in several previous studies, however, we feel that our study merits publication in Nature Communications due to the fact that

- (1) For such significant climatic effects on the global carbon sinks and sources, having multiple papers and research results are in fact important contributions that can in fact provide consistency and consensus among scientists and global climate policy institutions. Therefore, I think we should always welcome similar conclusions from different studies about such an important problem.
- (2) Our study is very different from previously published results based on ground plot data. Our paper documents a widespread and persistent impact of a severe drought on carbon dynamics of the Amazonian forests derived from satellite-based canopy structure measurements, which were missing from any other systematic measurements from space.
- (3) The results shown here are also different from other studies that only report changes from some scattered plots in the Amazonia. To quantify the widespread effect of droughts, we either need a systematic or probabilistic sampling approach from ground plots or the use of systematic remote sensing observations as shown in this paper. Therefore, our approach can be considered more spatially consistent over the entire Amazon.
- (4) The reported changes have undergone robust uncertainty analysis that considered all possible sources of error from AGB estimation, sampling errors, and mapping uncertainties.

Reviewer #3 (Remarks to the Author):

The authors have made considerable improvements to the ms, and it is now close to publishable quality. I have some comments that need to be addressed prior to this:

Author response: We thank the reviewer for giving our paper positive comments. We have also improved the manuscript following the reviewer's comments and suggestions.

“The decline of carbon stocks started from the hydraulic stress...”. This is an inference, because hydraulic stress is not directly measured. It would be better to add “We infer that...” at the start of this sentence.

Author response: We agree with the reviewer's comment, and have revised the sentence as recommended by the reviewer.

The text notes that inventory plots suggest the 2005 drought converted forests from a “sink of about 0.71 MgC ha⁻¹ yr⁻¹ to a net source of carbon to the atmosphere of about 1.5 MgC ha⁻¹ yr⁻¹”. I suggest that the units are converted to MgC ha⁻¹, to match the numbers from the introductory sentence, and because of the extrapolation challenges of these plot level data noted in the next sentence.

Author response: We thank the reviewer for pointing this out regarding the source of carbon in terms of loss or rate of loss. We have replaced the second number (1.5 MgC ha⁻¹ yr⁻¹) to be the reported total loss number (5.3 MgC ha⁻¹ for forest subjected to a 100-millimeter increase in water deficit) in literature. But the first number was reported as an average rate of carbon sink from the long-term monitoring of mature tropical forests as a pre-drought condition (Phillips, O. L. et al. Changes in the Carbon Balance of Tropical Forests: Evidence from Long-Term Plots. Science 282, 439–442 (1998)). Therefore, we decided not to modify this number.